# Glutamate indicators with increased sensitivity and tailored deactivation rates

Abhi Aggarwal [1,2,3,15], Adrian Negrean [1,15], Yang Chen [4,15],
Rishyashring Iyer [5,15], Daniel Reep [2,6], Anyi Liu[7], Anirudh Palutla [2],
Michael E. Xie [1,8], Bryan J. MacLennan[1], Kenta M. Hagihara[1], Lucas W. Kinsey[1,2],
Julianna L. Sun[9], Pantong Yao[10], Jihong Zheng [2,6], Arthur Tsang[2,6],
Getahun Tsegaye[2,6], Yonghai Zhang [4], Ronak H. Patel[2], Benjamin J. Arthur [2],
Julien Hiblot [11], Philipp Leippe[12], Miroslaw Tarnawski [11],
Jonathan S. Marvin [2], Jason D. Vevea[9], Srinivas C. Turaga [2], Alison G. Tebo [2],
Matteo Carandini [7], L. Federico Rossi [7,13], David Kleinfeld [5,14],
Arthur Konnerth [4], Karel Svoboda [1,2], Glenn C. Turner [2,6],
Jeremy P. Hasseman [2,6] ✉ & Kaspar Podgorski [1,2] ✉

Understanding how neurons integrate signals from thousands of input synapses requires methods to monitor neurotransmission across many sites simultaneously. The fluorescent protein glutamate indicator iGluSnFR enables visualization of synaptic signaling, but the sensitivity, scale and speed of such measurements are limited by existing variants. Here we developed two highly sensitive fourth-generation iGluSnFR variants with fast activation and tailored deactivation rates: iGluSnFR4f for tracking rapid dynamics, and iGluSnFR4s for recording from large populations of synapses. These indicators detect glutamate with high spatial specificity and single-vesicle sensitivity in vivo. We used them to record natural patterns of synaptic transmission across multiple experimental contexts in mice, including two-photon imaging in cortical layers 1–4 and hippocampal CA1, and photometry in the midbrain. The iGluSnFR4 variants extend the speed, sensitivity and scalability of glutamate imaging, enabling direct observation of information flow through neural networks in the intact brain.

Neurons process information by combining and transforming signals arriving at their many synaptic inputs[1–3]. A major goal of brain research is therefore to monitor the activity of input synapses and neuronal output in the intact brain. Fluorescent calcium indicators[4], voltage indicators[5–7] and extracellular electrophysiology[8] are commonly used to record outputs from large populations of neurons[9]. By contrast, there are no technologies for recording large populations of synaptic inputs.

The large majority of vertebrate central synapses release the neurotransmitter glutamate[3]. One action potential (AP) typically releases zero or one vesicle, freeing a few thousand glutamate molecules, which are cleared from the synaptic cleft in less than 1 ms[10]. In comparison, one AP triggers the influx of more than $1 \times 10^5$ calcium ions into a typical

neuronal cell body, which are cleared within 20 ms[11]. The small number of glutamate molecules and their short residence time make it challenging to obtain optical measurements of synaptic release, especially when recording from many synapses at once in vivo. Addressing this challenge is necessary to reveal fundamental principles of neuronal computation, such as whether the non-linearities that implement computations occur primarily in the soma or dendrites[1,12], and what patterns of synaptic input drive neuronal firing and plasticity in vivo[2]. Neurotransmitter recordings can also be used to study synaptic plasticity[13–17], neural connectivity[18], neurological disorders[19–21] and the molecular mechanisms and pharmacology of synaptic transmission[17,22–24].

Fluorescent protein neurotransmitter indicators are fusions between a binding domain and a fluorescent protein domain, displayed

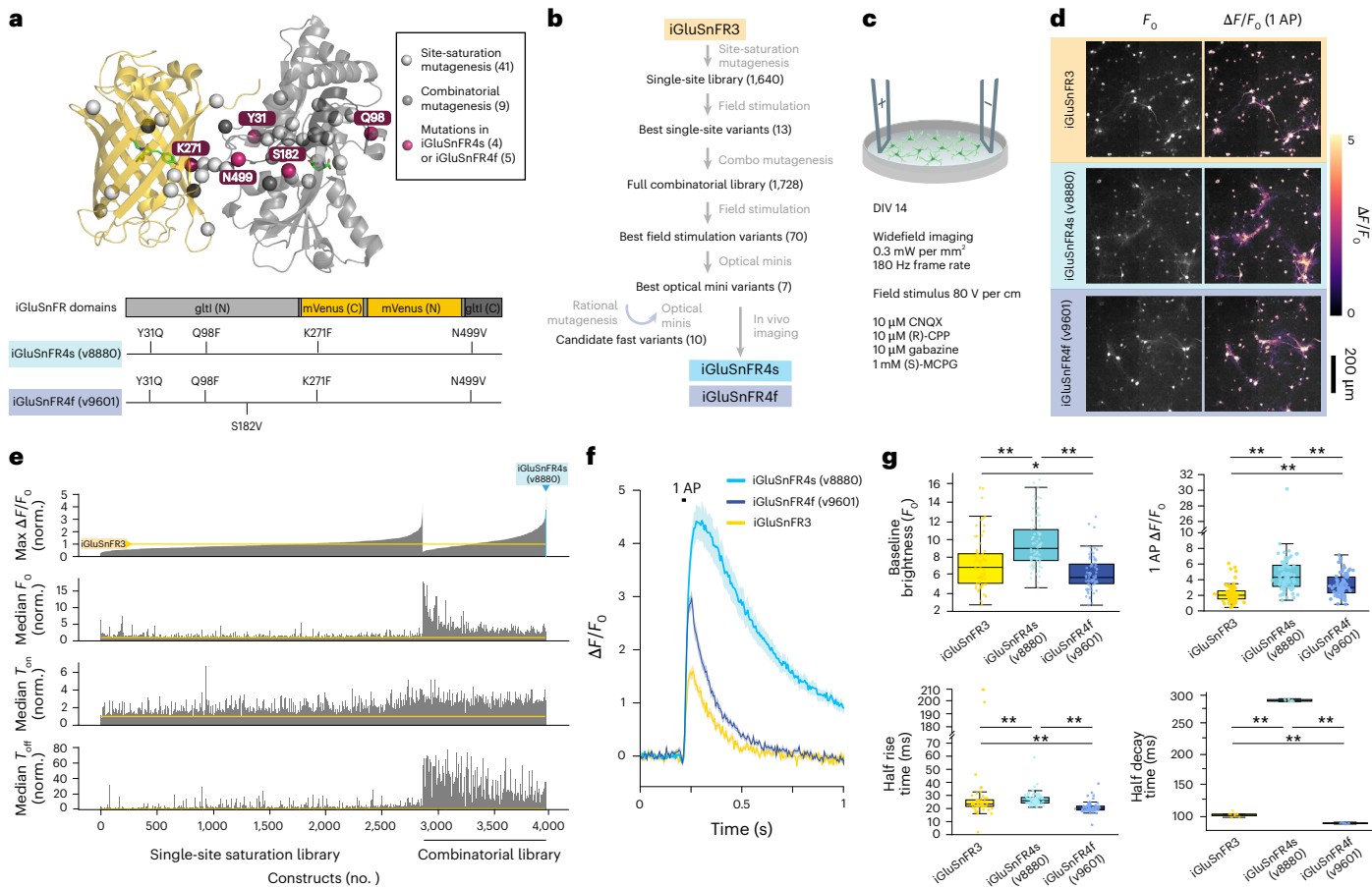

**Fig. 1 | Screening iGluSnFR variants in primary neuronal culture. a**, Top: location of mutagenesis sites, mapped onto the iGluSnFR3 structure. Light gray spheres, sites selected for saturation mutagenesis; dark gray spheres, sites included in full combinatorial mutagenesis; red spheres, sites included in iGluSnFR4s or iGluSnFR4f. Bottom: mutations in iGluSnFR4s and iGluSnFR4f. **b**, Screening steps. **c**, Primary neuron cultures expressing iGluSnFR variants received a brief electric field stimulus to evoke a single action potential. **d**, Images of cultures expressing iGluSnFR3, iGluSnFR4s and iGluSnFR4f at $F_0$ (left) and their peak $\Delta F/F_0$ (right) following a single field stimulus.

The experiment was repeated independently 66 times with similar results. **e**, Measured $\Delta F/F_0$, $F_0$, $T_{on}$ and $T_{off}$, normalized to within-plate iGluSnFR3 controls, ordered by $\Delta F/F_0$ for responsive constructs in the single-site and combinatorial libraries. $n = 4$–16 wells per variant. **f**, $\Delta F/F_0$ fluorescence traces (mean ± s.e.m.) in the field stimulation assay for iGluSnFR3, iGluSnFR4f and iGluSnFR4s. **g**, $F_0$, peak $\Delta F/F_0$, $T_{on}$ and $T_{off}$ of the three variants. Boxplots denote mean, interquartile range and extrema excluding outliers, with all individual measurements overlaid. *$P < 0.05$, **$P < 0.01$, two-sided Mann–Whitney $U$ test. $n$ (wells) = 66, 74 and 73.

on the cell surface[25]. Conformational changes upon ligand binding modulate the indicator fluorescence, which can be read out at high resolution with a microscope. The fluorescent protein glutamate indicator iGluSnFR can detect the release of individual synaptic vesicles under favorable conditions[13,15,22]. However, even for the variant with the highest signal-to-noise ratio (SNR), iGluSnFR3 (ref. 26), the SNR in vivo is often insufficient to record from more than a few dozen synapses at once. The SNR decreases when more synapses are imaged because limited total excitation power must be divided across synapses[27,28]. Brighter, more sensitive indicators are required for measurements from larger groups of individual synapses.

Additional constraints arise from the finite voxel rates of microscopes, imposing a trade-off between the sampling rate and the number of recorded synapses. For example, conventional two-photon microscopy can address only a few dozen synapses at the ~100-Hz frame rate required to resolve iGluSnFR3 glutamate transients, which decay with a time constant below 30 ms[26]. Variants with slower deactivation kinetics facilitate imaging at lower frame rates, allowing access to more synapses. By contrast, variants with fast deactivation kinetics would allow more precise monitoring of rapid synaptic dynamics[29,30], albeit limited to smaller imaging volumes. Regardless of deactivation rate, fast activation kinetics are needed for precise measurements of glutamate release times[27,31].

Here we engineered two highly sensitive, fast-activating iGluSnFR variants with fast or slow deactivation. We screened a large library of rationally targeted mutations in vitro for improved brightness, kinetics and sensitivity. We further tested seven variants using two-photon imaging in the mouse visual cortex. We selected two variants, iGluSnFR4f (fast deactivation) and iGluSnFR4s (slow deactivation), with substantially improved sensitivity and brightness. We highlight their advantages in experiments involving synaptic imaging in the visual cortex, somatosensory cortex and hippocampus, and fiber photometry in the midbrain.

## Results

### Cultured neuron field stimulation screen

Previous research has identified many sites that influence iGluSnFR function[26,29,30,32], but the large combinatorial sequence space across these sites has not been explored well. We conducted saturating mutagenesis at 41 previously identified sites in 2 iGluSnFR3 variants (iGluSnFR3.v857 and iGluSnFR3.v867), generating a total of 1,640 variants (Fig. 1a,b). We transduced these into primary cortical cultures from rats (14 days in vitro (DIV)), and imaged responses to field stimulation[33] (Fig. 1c,d). Baseline fluorescence brightness ($F_0$), peak fractional response ($\Delta F/F_0$) and rise and decay times ($T_{on}$ and $T_{off}$) were measured (Fig. 1e–g and Extended Data Fig. 1).

From this screen, we selected 12 point substitutions across 9 sites (p.Y31Q, p.Y31E, p.Q34A, p.Q98F, p.A185N, p.T254R, p.K271F, p.K271G, p.H273E, p.Q418S, p.N49L and p.N499V) that exhibited superior performance in at least one parameter ($F_0$, $\Delta F/F_0$, $T_{on}$ and/or $T_{off}$). We then constructed a complete combinatorial library comprising 1,728 variants on the iGluSnFR3.v857 background (hereafter referred to as iGluSnFR3). Of these, 1,392 variants were successfully expressed, exhibited detectable responses to field stimuli and were included in further analyses.

We used a generalized linear model (GLM) to quantify the effects of each substitution and their interactions (Extended Data Fig. 1a). Incorporating pairwise interactions into the model enhanced the cross-validated variance explained for all response variables, indicating that epistatic interactions were present (Extended Data Fig. 1b–e). We resolved a high-resolution crystal structure of iGluSnFR3 (PDB: 9FBU), allowing us to map substitution effects and interactions to physical positions. The magnitude of pairwise interactions was inversely correlated with distance ($r = -0.28$, $P = 5.34 \times 10^{-6}$; Extended Data Fig. 1f). Although local in physical space, many of the strongest interactions were between residues from different domains of the fusion protein's linear sequence (Extended Data Fig. 1g).

### Optical mini screen

We selected 70 variants from the combinatorial screen and tested them by imaging spontaneous synaptic glutamate release ('optical minis') in cultured neurons silenced with tetrodotoxin (TTX). Optical minis are thought to reflect the asynchronous release of individual synaptic vesicles[22,26,34,35] and have a much smaller spatial span, and therefore faster diffusion-limited kinetics, than do field-evoked responses (Fig. 2a–c). Optical mini imaging is thus well-suited to identify variants with fast decay kinetics and improved SNRs for synaptic glutamate dynamics. Many variants exhibited higher SNRs than that of iGluSnFR3, with the top-performing variant, v8880 (iGluSnFR3-Y31Q Q98F K271F N499V), having a 4.7-fold-higher SNR and 1.2-fold-slower decay rate (Fig. 2d and Supplementary Video 1). Variants in the combinatorial screen and mini screens had slower-decaying signals than that of iGluSnFR3 (Figs. 1e and 2d), suggesting that the field-stimulation screen favored slower indicators. To create high-SNR variants with fast decay kinetics, we introduced five additional substitutions linked to short decay rates (p.S70A, p.S70T, p.S182L, p.S182V or p.Y209F)[29,30,32,36] onto two high-sensitivity variants, v8880 and v8376. These ten variants (v9598 through v9607) were further characterized with optical mini recordings. The best-performing of these, v9601 (v8880 + A182V), exhibited a 2.1-fold higher SNR and 3.2-fold faster decay than those of iGluSnFR3 (Fig. 2d and Supplementary Video 1).

### Mouse visual cortex screen and selection of iGluSnFR4s and iGluSnFR4f

We selected seven variants for in vivo testing (Fig. 2e–i) using the Nogo Receptor (NGR) signal sequence[26,37] for membrane display. We sparsely labeled neurons in the primary visual cortex (V1) through the coinfection of adeno-associated virus (AAV)-expressing Cre-dependent iGluSnFR and low titers of AAV-expressing Cre. First, we imaged the dendrites of pyramidal neurons in layer 2/3 while we presented periodic full-field light-flash visual stimuli (1 s on, 1 s off). Fluorescence transients were extracted using an algorithm based on non-negative matrix factorization. Amplitude SNR, detectability (a SNR measure that accounts for transient duration), $F_0$ and photobleaching were quantified using a GLM to control for imaging depth and expression time (Fig. 2i–l). Photobleaching was highly correlated with $F_0$ ($r^2 = 0.86$, $P = 2.9 \times 10^{-4}$), suggesting that variations in bleaching across variants were due largely to differences in the per-molecule excitation rate at baseline. v8880 and v9601 showed the highest SNR among the tested variants, consistent with the optical mini screen. Relative transient durations in vivo aligned with the results of the optical mini screen, revealing that v9601 was the fastest variant tested.

Despite shorter transients, detectability of v9601 was higher than that of iGluSnFR3.

Dendritic spines are specialized protrusions that receive the majority of glutamatergic inputs onto pyramidal neurons. Their addition, retraction and morphological changes are regulated by activity[38]. We tested whether expression of iGluSnFR variants affects spine survival. We did not see differences between iGluSnFR-expressing neurons and neurons expressing membrane-tagged eGFP (Supplementary Fig. 1).

On the basis of these assays, we selected v8880 and v9601 as the best-performing slow- and fast-decay variants, respectively, naming them iGluSnFR4s and iGluSnFR4f. We characterized both variants in purified soluble protein (Extended Data Table 1, Extended Data Figs. 2 and 3 and Supplementary Figs. 2 and 3) and on cultured neurons (Extended Data Figs. 4 and 5). Purified protein measurements revealed strong modulation of both extinction and fluorescence quantum yield upon glutamate binding and high selectivity over other neurotransmitters and amino acids, except for aspartate, similar to previous iGluSnFRs. In neuronal culture, the single-AP $\Delta F/F_0$ of iGluSnFR4s and iGluSnFR4f was higher than that of iGluSnFR3 (Fig. 1d–g); additionally, the on-membrane affinity of iGluSnFR4s surpassed that of iGluSnFR3, whereas that of iGluSnFR4f was lower (Extended Data Fig. 4). iGluSnFR4s has a similar rise to that of iGluSnFR3 and much slower decay, whereas iGluSnFR4f exhibits faster rise and decay (Fig. 1i). As a result, iGluSnFR4f better follows rapid synaptic release in culture than does iGluSnFR3 (Extended Data Fig. 5).

### iGluSnFR4 detects synaptic glutamate in the visual cortex

To assess in vivo performance, we first expressed iGluSnFR3, iGluSnFR4s and iGluSnFR4f sparsely in neurons in V1 using single-cell electroporation. We then used loose cell-attached recording and two-photon imaging (500 Hz, <30 mW) to characterize single-AP-evoked glutamate transients on axonal boutons of the recorded neurons (Fig. 3a,b). All three indicators exhibited fluorescence transients time-locked with somatically recorded APs (Fig. 3b,c). We computed spike-triggered averages (STAs) (Fig. 3c), which were well described by a fast rise and slower exponential decay. iGluSnFR4f and iGluSnFR4s showed substantially larger single-AP-evoked amplitudes than did iGluSnFR3 (Fig. 3d). For all three indicators, rise times were less than 2 ms (Fig. 3d). The decay time constant of iGluSnFR4f was faster (median, 25.9 ms; interquartile range (IQR), 24.2–27.6 ms), and that of iGluSnFR4s was much slower (median, 152.7 ms; IQR, 134.–9163.5 ms), than that of iGluSnFR3 (median, 29.1 ms; IQR, 25.6–32.6 ms) (Fig. 3d). In each of the iGluSnFR4 variants, single-AP detectability was improved over that of iGluSnFR3 (Extended Data Fig. 6). To highlight the high brightness and sensitivity of iGluSnFR4f, we imaged activity in hippocampal CA1 dendrites, including of apical dendrites >480 μm below the surface of an implanted optical cannula (Extended Data Fig. 7).

Next, we compared the ability of the iGluSnFR variants to resolve signals at individual synapses. Cortical neuropil contains approximately one glutamatergic synapse per cubic micrometer[39,40], requiring that indicators show high spatial specificity to avoid cross-talk. The kinetic properties of indicators shape the spatial extent of fluorescence signals[26]. Indicators with slower kinetics or that saturate near the site of release, where glutamate concentrations are high, result in signals with larger spatial extent.

To assess spatial extent and cross-talk, we imaged dendritic responses to directional drifting grating stimuli in V1 (Fig. 3e). Synaptic inputs to V1 neurons have diverse preferences for grating orientation, direction and phase, exhibiting little spatial organization over micrometer scales[41]. Spatial cross-talk would blend signals with different tuning, reducing measured selectivity. Both iGluSnFR4f and iGluSnFR4s exhibited high-SNR, orientation-tuned responses that were localized to dendritic spines (Fig. 3f–j). We observed clear responses for each cycle of the grating stimuli from many distinct sites. iGluSnFR4f responses were sharper in time (Fig. 3f–h), whereas iGluSnFR4s responses have larger

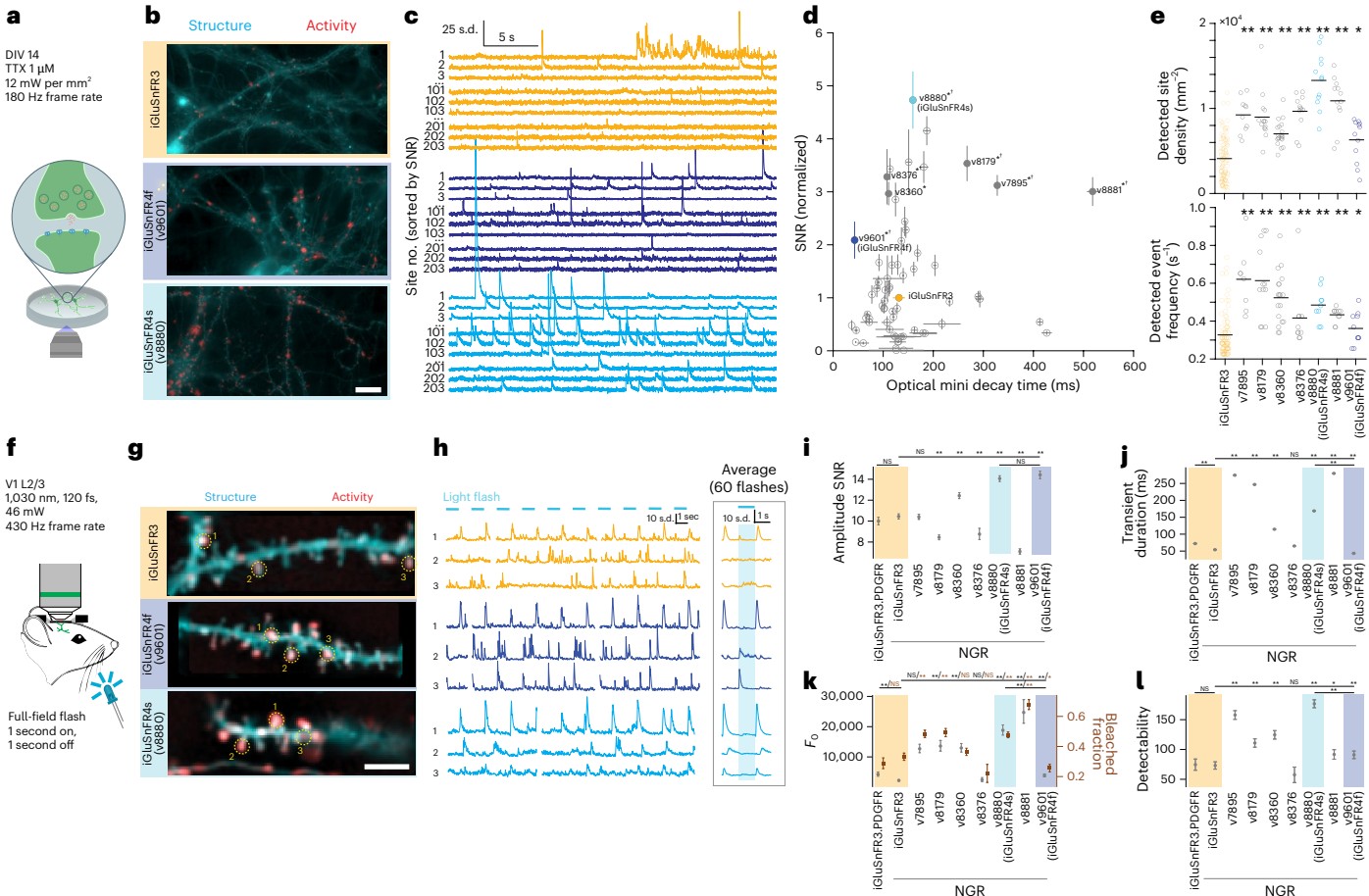

**Fig. 2 | Synapse-resolved screening in cultured neurons and the visual cortex in vivo. a**, iGluSnFR variants underwent high-speed, high-resolution widefield imaging of spontaneous events (optical minis) in primary neuronal cultures silenced with TTX. **b**, Representative structural (cyan; gamma 0.5) and pixel-wise activity (normalized skewness, red) images for iGluSnFR3, v9601 (iGluSnFR4f) and v8880 (iGluSnFR4s), on matching color scales across panels for both channels. Scale bar, 20 μm. The experiment was repeated independently 12 times, with similar results. **c**, $\Delta F$ traces for individual sites for each variant, each normalized to its standard deviation (s.d.), and ordered by SNR. Sites ranked 1–3, 101–103 and 201–203 are shown. **d**, The SNR and decay time (mean ± s.e.m. across wells; $n$ = 99 wells (iGluSnFR3), 10–20 wells (others)) of variants in optical mini screen. Filled circles denote variants selected for in vivo characterization. [†]$P$ < 0.05 decay time versus iGluSnFR3; *$P$ < 0.05 SNR versus iGluSnFR3. Two-sided Mann–Whitney $U$ test. **e**, Density of detected sites (top) and median mini frequency per site (bottom) for the variants selected for in vivo testing. $n$ (wells) = 99 iGluSnFR3, 10 v7895, 14 v8179, 20 v8360, 12 v8376, 12 v8880, 12 v8881 and 12 v9601. Black lines denote mean over wells of each variant. *$P$ < 0.05,**$P$ < 0.01, two-sided Mann–Whitney $U$ test versus iGluSnFR3. **f**, Dendrites of sparsely

labeled layer 2/3 neurons in V1 were recorded with 2P imaging while a full-field light flash stimulus was presented. **g**, Representative structural images (cyan; gamma 0.5) and pixel-wise activity (red) for iGluSnFR3, v9601 (iGluSnFR4f) and v8880 (iGluSnFR4s), on matching color scales across panels for both channels. Circles labeled 1–3 denote regions of interest (ROIs) for corresponding traces in **h**. Scale bar, 10 μm. The experiment was repeated independently 15 times with similar results. **h**, Representative traces (left) and stimulus-triggered averages (right) for ROIs in **g**. Gaps in traces correspond to frames discarded owing to movement. **i–l**, Summaries of SNR (**i**), transient duration (**j**), $F_0$ and bleached fraction (brown markers, larger values denote more bleaching) (**k**) and detectability (**l**) for the screened variants. Shaded colors highlight iGluSnFR3 and the selected iGluSnFR4 variants. Except for transient duration, which is reported directly, data points and error bars are GLM-estimated marginal means and standard error. Confidence intervals (CIs) account for covariates of expression time and imaging depth. $n$ (distinct branches) = 15 iGluSnFR3.PDGFR, 52 iGluSnFR3, 19 v7895, 17 v8179, 31 v8360, 40 v8880, 12 v8881 and 25 v9601. NS, not significant, *$P$ < 0.05,**$P$ < 0.01, two-sided Mann–Whitney $U$ (transient duration) or two-sided $t$-test (others).

peak amplitudes and integrated areas under the curve (Fig. 3i). The percentage of tuned responses reported by iGluSnFR4s was lower than that of iGluSnFR4f (Fig. 3j), suggesting slightly more cross-talk with the slower, higher-affinity indicator. We next measured the spatial extent of signals using cross-trial tuning covariances. Because the tuning of nearby synapses is largely uncorrelated, the measured covariance in tuning between pairs of pixels (calculated from non-overlapping sets of trials) reflects the spatial spread of signals arising from a synapse[42]. iGluSnFR4f showed a slightly narrower covariance than did iGluSn-FR4s, although neither differed significantly from that of iGluSnFR3 ($P$ = 0.075 iGluSnFR4f versus iGluSnFR3, $P$ = 0.552 iGluSnFR4s versus iGluSnFR3, bootstrap test; Fig. 3k). These results show that, similar to iGluSnFR3, iGluSnFR4f and iGluSnFR4s report signals from individual

synapses in vivo with high specificity, although iGluSnFR4f has slightly higher specificity than does iGluSnFR4s.

To assess the photostability of the new indicators, we recorded the dendrites of virally labeled neurons in V1 continuously for 1 h at 100 Hz using 6.9 μW μm⁻² of excitation power while mice were presented with drifting gratings (Extended Data Fig. 8). Synapses retained their characteristic highly tuned responses and high signal levels (87 ± 8.4% (iGluSnFR4s) and 75 ± 10% (iGluSnFR4f) of initial; mean ± s.e.m.) at the end of the 1 h continuous high-speed recording.

**Frequency response characterization in barrel cortex layer 4**
Rodents sense the world by rhythmically sweeping their vibrissae (whiskers) over objects at around 15 Hz[43]. Rapidly varying signals from

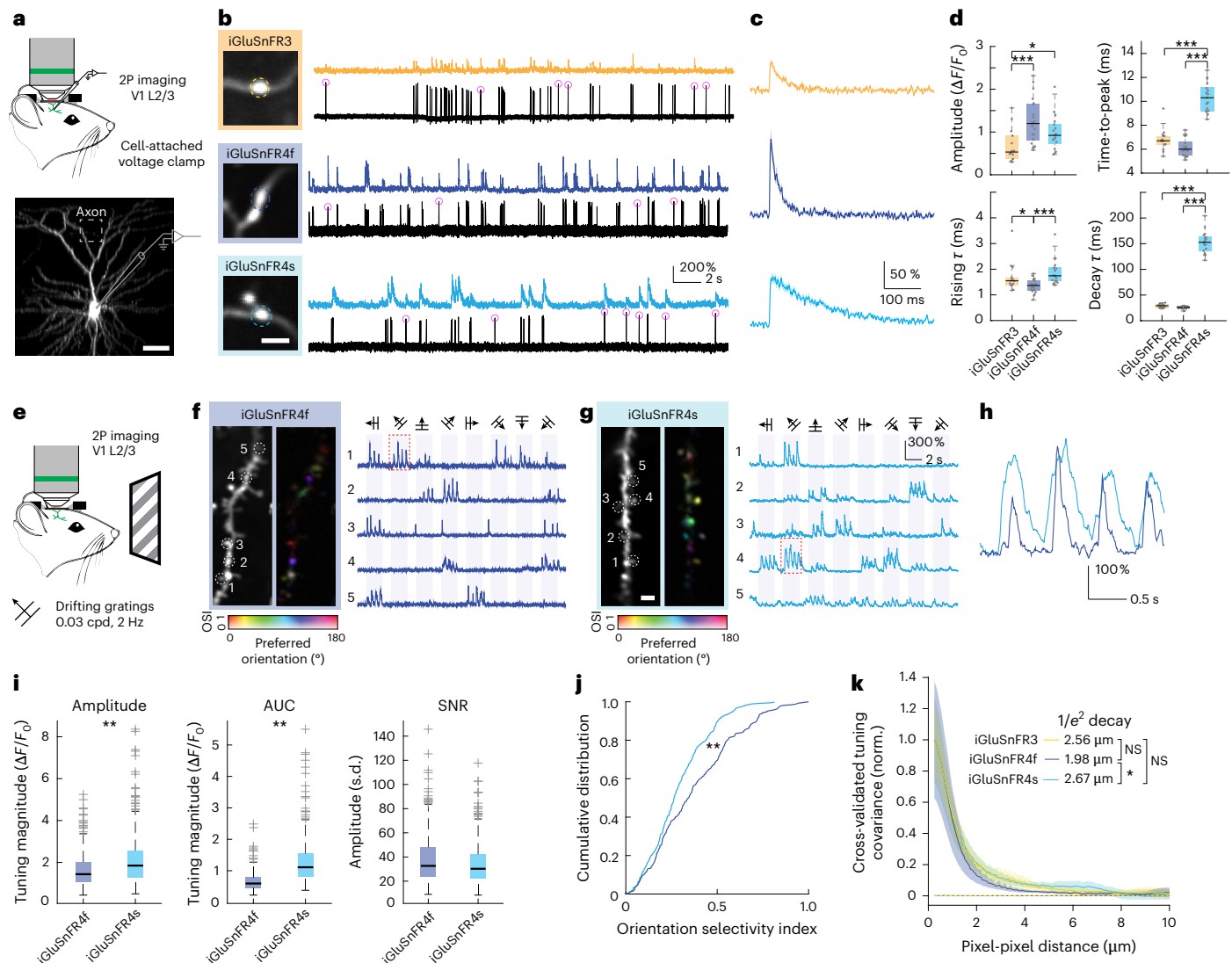

**Fig. 3 | Characterization of iGluSnFR4 in the mouse visual cortex. a**, Layer 2/3 neurons in V1 were transduced with plasmids expressing iGluSnFR variants using single-cell electroporation. Following expression, simultaneous axonal imaging and loose-seal, cell-attached recordings were performed. Scale bar, 50 µm. **b**, Axonal glutamate signals and somatic spiking for neurons expressing iGluSnFR3, iGluSnFR4f and iGluSnFR4s. Pink circles indicate single APs. Scale bar, 2 µm. In **a** and **b**, the experiment was repeated independently four times, with similar results. **c**, Spike-triggered averages for 1AP-evoked axonal glutamate signals. Shading denotes the s.e.m. $n = 46$ iGluSnFR3, 24 iGluSnFR4f and 58 iGluSnFR4s isolated spikes. **d**, Amplitude, time-to-peak, rise and decay time constants ($\tau$) for the 1AP glutamate transients from iGluSnFR3 ($n = 14$ boutons from 4 cells), iGluSnFR4f ($n = 21$ boutons from 4 cells) and iGluSnFR4s ($n = 17$ boutons from 4 cells). Kruskal–Wallis test with Dunn's test for multiple comparisons. **e**, Layer 2/3 dendrites were imaged while stimulating the contralateral eye with oriented drifting gratings. **f**, Two-photon image of a dendritic segment from an iGluSnFR4f-labeled neuron in layer 2/3 and a pixelwise tuning map (left). Representative single-trial $\Delta F/F_0$ traces from denoted ROI are shown on the right. **g**, The same as in **f** but for iGluSnFR4s. Scale bar, 2 µm. In **f** and **g**, the experiment was repeated independently four times with similar results. **h**, Expanded traces from the red boxes. **i**, Tuning magnitude

for iGluSnFR4f and iGluSnFR4s computed from the response amplitude (left, $n = 324$ ROIs for iGluSnFR4f and $n = 349$ ROIs for iGluSnFR4s from 4 cells each, $P = 9.6 \times 10^{-9}$, two-sided Wilcoxon rank sum test) or the area under the curve (AUC) (middle, $n = 267$ ROIs for iGluSnFR4f and $n = 348$ for iGluSnFR4s from 4 cells each. $P = 8.4 \times 10^{-49}$, two-sided Wilcoxon rank sum test). Right: SNRs of responses to the preferred direction at each site ($n = 324$ ROIs for iGluSnFR4f and $n = 349$ ROIs for iGluSnFR4s from 4 cells each, $P = 0.052$, two-sided Wilcoxon rank sum test). **j**, Distribution of orientation selectivity index for iGluSnFR4s and iGluSnFR4f ($n = 267$ ROIs for iGluSnFR4f and $n = 348$ ROIs for iGluSnFR4s from 4 cells each, $P = 4.8 \times 10^{-6}$, two-sided Kolmogorov–Smirnov test). **k**, Cross-validated covariances for pairs of labeled pixels at various distances, indicative of the spatial spread of iGluSnFR signals around release sites, for iGluSnFR3, iGluSnFR4s and iGluSnFR4f. Dashed curves denote covariances calculated over unlabeled pixels. $n = 8$ fields of view per variant. Error bars denote bootstrapped 95% CIs. Mean distances to $1/e^2$ of peak labeled pixel covariance are shown. $P = 0.075$ iGluSnFR4f versus iGluSnFR3, $P = 0.552$ iGluSnFR4s versus iGluSnFR3, $P = 0.005$ iGluSnFR4s versus iGluSnFR4f, bootstrap test. In **d** and **i**, boxplots denote median and interquartile range; outliers beyond 1.5 × interquartile range are plotted separately. *$P < 0.05$, **$P < 0.01$, ***$P < 0.001$.

the vibrissa follicles ascend to the primary somatosensory cortex, vS1, through the ventral posteromedial (VPM) thalamus. To characterize stimulus-evoked glutamatergic signals across frequencies, we expressed iGluSnFR3, iGluSnFR4s and iGluSnFR4f by injecting AAVs into VPM, then recorded from thalamocortical axons 350–400 µm deep

in vS1. Imaging was done with an adaptive-optics two-photon (AO-2P) microscope in head-fixed mice during rhythmic air-puff stimulation (Fig. 4a–d) and voluntary whisking (Fig. 4e–i).

We observed rapid-onset responses, time-locked to rhythmic stimulation ranging from 2–30 Hz across all indicators (809 boutons,

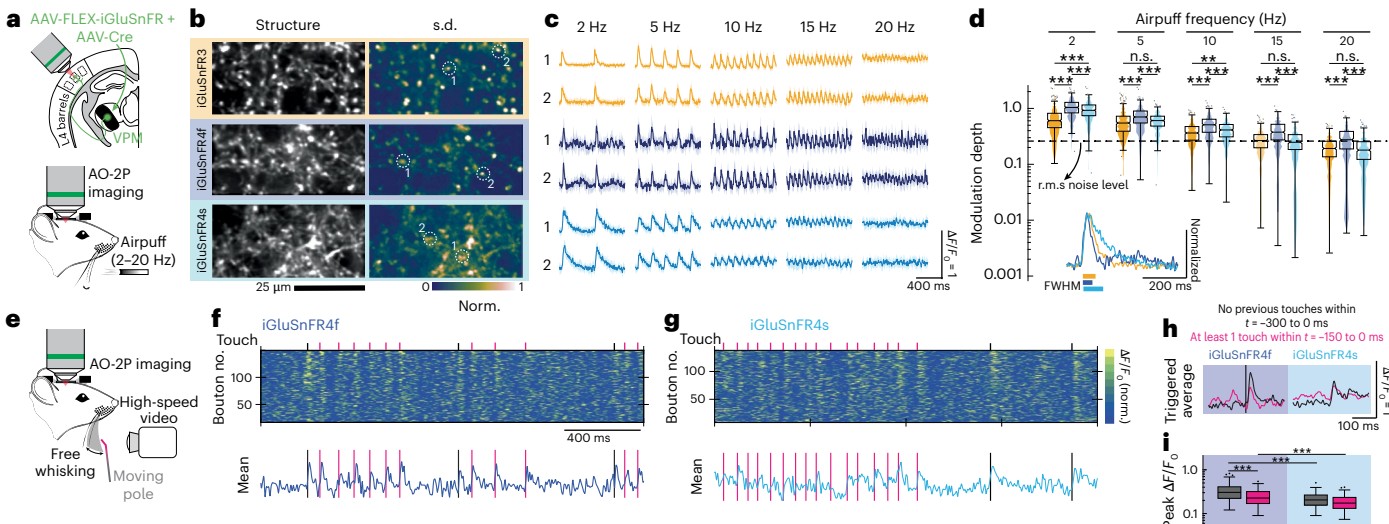

**Fig. 4 | Imaging rapid dynamics in thalamocortical axons using AO-2P scanning microscopy. a**, iGluSnFR variants were expressed by AAV injection in VPM and thalamocortical boutons in the barrel cortex, recorded with AO-2P imaging. Awake mice received rhythmic airpuff stimulation to their vibrissae during high-speed AO-2P imaging. **b**, Left: average image of thalamocortical axons in L4 labeled by the iGluSnFR variants. Right: normalized pixelwise s.d. across 1-second averaged epochs from 60 trials under the 5 Hz airpuff stimulus. The experiment was repeated independently two (iGluSnFR4f) or three (others) times, with similar results. **c**, Responses to the indicated airpuff frequencies for boutons marked in **b**; the mean with the interquartile range ($n = 20$ trials) is shown. **d**, A comparison of iGluSnFR variant response amplitudes in thalamocortical boutons at different stimulation frequencies during a 4-second airpuff stimulus: iGluSnFR3 (3 mice, 431 boutons), iGluSnFR4f (2 mice, 407 boutons) and iGluSnFR4s (3 mice, 409 boutons). The dashed line indicates root mean square (r.m.s.) noise level for iGluSnFR3. ***$P < 0.0001$, **$P < 0.001$; two-sided two-sample $t$-test with unequal variances. Inset: normalized airpuff-triggered average for a single bouton for each of the iGluSnFR variants, mean

over 100 stimulus pulses. The extent above half-maximum is plotted below. FWHM, full width at half maximum. **e**, Awake mice were allowed to whisk freely against a moving pole during high-speed AO-2P imaging of thalamocortical boutons and high-speed videography to record touch events. **f**, Top: $\Delta F/F_0$ traces of individual boutons. Bottom: average $\Delta F/F_0$ traces of the 10 most responsive boutons over a 2-second period from a single field labeled with iGluSnFR4f. The black and pink lines mark the instances of the two classes of pole touch events in **i**. **g**, The same as in **f**, but for iGluSnFR4s. **h**, Touch-triggered mean responses. Black curves indicate responses from touch events with no preceding touches within 300 ms (21 events for iGluSnFR4f, 27 events for iGluSnFR4s). Pink curves indicate responses with at least 1 touch event in the preceding 150 ms (67 events for iGluSnFR4f, 88 events for iGluSnFR4s). **i**, Distribution of peak responses from the two categories, for all boutons (iGluSnFR4f, 2 mice, 127 boutons; iGluSnFR4s, 2 mice, 148 boutons). From left to right: $P = 1.1462 \times 10^{-6}$, $P = 5.2014 \times 10^{-11}$; $P = 9.8452 \times 10^{-7}$; two-sided two-sample $t$-test with unequal variances. In **d** and **i**, boxplots denote the median and interquartile range; outliers beyond $1.5 \times$ interquartile range are plotted separately.

7 mice) (Fig. 4c and Supplementary Video 2). The modulation depth of iGluSnFR signals decreased with increasing stimulus frequency, aligning with each variant's finite response bandwidth (Fig. 4c). iGluSnFR4f exhibited the highest modulation across all frequencies (Fig. 4d) and maintained detectable responses with stimulation up to 20 Hz, exceeding the natural whisking frequency of mice.

We next characterized the responses of iGluSnFR4 to touch while mice were free-whisking against a pole; touch events were recorded with a high-speed camera (Fig. 4e). The pole was moved randomly to add variability to inter-touch intervals. We compared two categories of touch events: ones with no touches in the preceding 300 ms, and ones with at least one touch in the preceding 150 ms. Although iGluSnFR4s consistently captured the former category, responses to rapidly recurring touch events were poorly resolved owing to its slow decay time; iGluSnFR4f produced resolvable responses for both categories (Fig. 4f,g). On average, iGluSnFR4f produced higher-amplitude responses for both categories (Fig. 4h,i), making it well-suited for monitoring inputs to the somatosensory cortex in rodents.

### Imaging iGluSnFR4s at slower frame rates in tuft dendrites of layer 5 neurons

In two-photon imaging, limitations on scanner line rates and pixel dwell time create tradeoffs between frame rate and imaging volume. For example, resonant scanning of a 512-line image is limited to ~30 Hz, too slow to adequately sample iGluSnFR3 transients. We evaluated the utility of iGluSnFR4s for large-scale 30-Hz recordings by imaging tuft

dendrites of AAV-transfected layer 5 neurons in mouse V1 during presentation of sparse noise stimuli (Extended Data Fig. 9a). We processed these recordings with an automated processing pipeline (Suite2p[44]) to segment synaptic sites. For each synaptic site, we estimated the region of the visual field that drove visual responses by fitting a cross-validated spatiotemporal receptive field (stRF) (Extended Data Fig. 9b–g). The stRFs obtained with iGluSnFR4 explained four times more variance than did those derived from iGluSnFR3 (medians of 18.3% versus 4.5%, Extended Data Fig. 9j–l). We defined responsive sites as those with >10% of variance explained by their stRF and more variance explained than by chance. iGluSnFR4s yielded more than nine times more responsive sites per dendrite than did iGluSnFR3 (medians of 19% versus 2%, Extended Data Fig. 9m).

### High-sensitivity photometry with iGluSnFR4s

Fiber photometry, which involves delivering excitation light and collecting the resulting fluorescence through an optical fiber without forming an image, has been broadly adopted for neuroscience measurements in deep regions[45,46]. Because photometry integrates signals from all labeled membranes, it benefits from high-affinity indicators, which capture more ligand farther from release sites. Photometry also benefits from slower indicators because these generate more time-integrated photons per release event. Because the large volumes sampled in photometry recordings average signals across neurons, the resulting signals are slower, reducing the benefits of very fast indicators. For these reasons, iGluSnFR3 shows similar-amplitude photometry responses to the slower, higher-affinity variant SF-iGluSnFR.A184S[46], despite producing

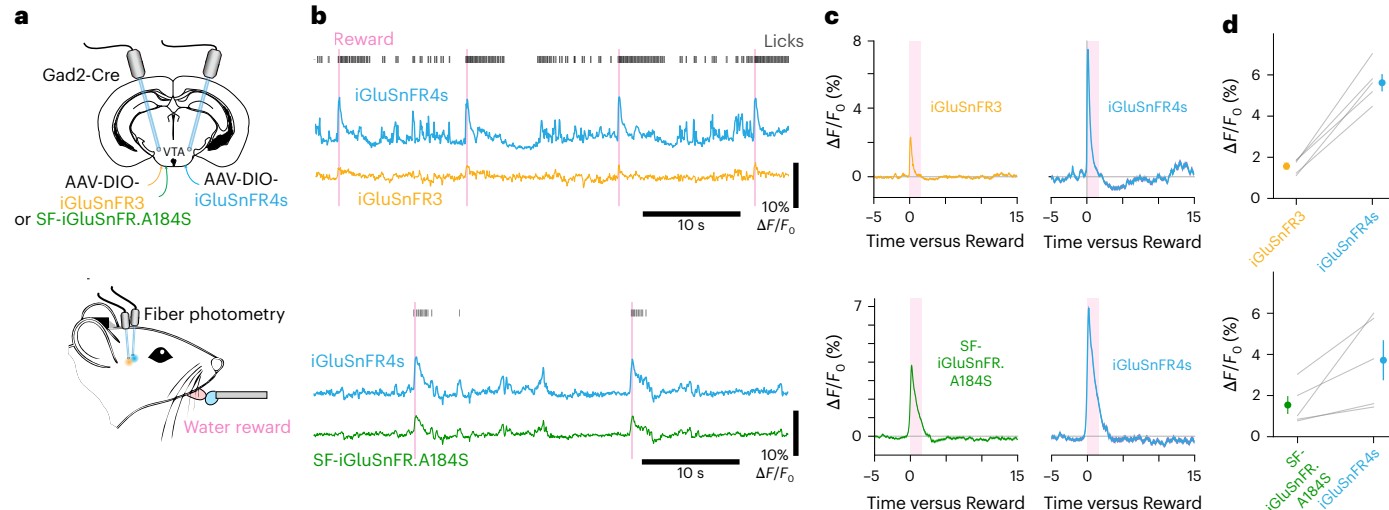

**Fig. 5 | Deep-brain fiber photometry with iGluSnFR4s. a**, iGluSnFR4s and either iGluSnFR3 or SF-iGluSnFR.A184S were expressed virally, and fibers implanted, in the ventral tegmental areas of opposite hemispheres. Photometry from both hemispheres was performed in mice receiving periodic water rewards. **b**, Example recordings demonstrating responses at stimulus time and during licks. **c**, Trial averaged $\Delta F/F_0$ traces (mean ± s.e.m.) of the three indicators aligned to reward delivery. Shaded region shows the time window used to quantify mean $\Delta F/F_0$ in **d**. **d**, Mean and s.e.m. of the post-reward $\Delta F/F_0$ for the paired-hemisphere recordings. $P = 4.0 \times 10^{-4}$ (iGluSnFR3 versus iGluSnFR4s), $n = 5$ mice; $P = 0.0516$ (SF versus iGluSnFR4s), $n = 5$ mice; two-sided paired $t$-tests.

much larger and faster signals in high-resolution imaging[26]. We reasoned, therefore, that the slower signals, high affinity and high sensitivity of iGluSnFR4s could be especially well-suited for photometry. To test this, we expressed iGluSnFR4s and either iGluSnFR3 or SF-iGluSnFR in ventral tegmental area (VTA) GABAergic neurons, paired across brain hemispheres (Fig. 5a and Supplementary Fig. 4). We recorded from both hemispheres while presenting water-restricted mice with water rewards (Fig. 5b). All three indicators showed transients tightly coupled to rewards and licking, synchronized across hemispheres. In response to rewards, iGluSnFR4s responded with larger amplitude than did iGluSnFR3 (5.62 ± 0.42 versus 1.56 ± 0.16, mean ± s.e.m.) or SF-iGluSnFR (3.72 ± 0.92 versus 1.54 ± 0.44, mean ± s.e.m.) in the paired measurements (Fig. 5c,d), demonstrating its high performance in photometry applications.

## Discussion

Here we developed iGluSnFR variants with improved sensitivity and faster (iGluSnFR4f, 26 ms) or slower (iGluSnFR4s, 153 ms) deactivation times. We demonstrated the performance of iGluSnFR4f and iGluSnFR4s in a variety of applications, including in vivo recordings of axons and dendrites, and cell types from layers 2–5 of the neocortex, thalamus and hippocampus, as well as GABAergic midbrain neurons. As with calcium indicators[5,47], fast- and slow- deactivating iGluSnFRs each have distinct and valuable applications. iGluSnFR4f is best suited for following rapid inter-release-intervals and shows higher spatial specificity. iGluSnFR4s is best suited for detection of release owing to larger time-integrated photon counts and SNRs in two-photon recordings, greatly increased signals under slower (30 Hz) imaging and greatly increased signals in fiber photometry. iGluSnFR4s shows slower decay but a fast rise time (<2 ms), which can enhance detectability while allowing precise timing inference from transients' rising phase[31]. The increased SNR is critical for observing single-vesicle release[13,15,22,26], recording from larger synaptic populations[48] and other challenging applications. Recording from larger populations is needed to study the potentially complex but poorly understood input–output operations of individual neurons, which integrate input from thousands of excitatory synapses on millisecond timescales[1,2].

iGluSnFR4 was developed using neuron culture and in vivo assays to select for responses to individual release events, such as 1 AP field

stimulation and optical mini imaging. The relative performance of tested variants was largely consistent across assays, but we observed some differences. The slower rise and decay times observed in culture suggest that in vivo assays are crucial for evaluating in vivo performance. Both iGluSnFR4 variants exhibit a reduced dynamic range in response to saturating ligand in soluble protein, a commonly used screening assay. Low glutamate concentrations used in culture imaging likely favored high-affinity variants (such as iGluSnFR4s) during in vitro screening compared with in vivo applications. These differences underscore the importance of screening with assays that align with the intended applications.

Our combinatorial screen densely sampled application-relevant functional properties over the high-dimensional sequence space of 12 mutations, each exhibiting strong individual effects. We uncovered numerous non-linear interactions that could be useful for indicator engineering beyond this work, for example to train and test models that predict complex relationships between protein sequence and function.

We performed tests under both the PDGFR and NGR membrane display vectors. The NGR vector matched or outperformed PDGFR under side-by-side tests in vivo, and greatly outperformed PDGFR in culture[26,37]. We recommend the NGR vector for most applications.

In conclusion, iGluSnFR4s and iGluSnFR4f address critical challenges for in vivo recordings of populations of individual synaptic inputs and rapid synaptic dynamics. These tools will enable new studies in neuronal computation, synaptic physiology and mechanisms underlying brain disorders.

## Online content

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

¹Allen Institute for Neural Dynamics, Seattle, WA, USA. ²Janelia Research Campus, Howard Hughes Medical Institute, Ashburn, VA, USA. ³University of Calgary Cumming School of Medicine and Hotchkiss Brain Institute, Calgary, Alberta, Canada. ⁴Institute of Neuroscience and Munich Cluster for Systems Neurology, Technical University of Munich, Munich, Germany. ⁵Department of Physics, University of California, San Diego, La Jolla, CA, USA. ⁶The GENIE Project Team, Janelia Research Campus, Ashburn, VA, USA. ⁷University College London, London, UK. ⁸Johns Hopkins University, Baltimore, MD, USA. ⁹Neuronal Cell Biology Division, Department of Developmental Neurobiology, St. Jude Children's Research Hospital, Memphis, TN, USA. ¹⁰Department of Neurosciences, University of California, San Diego, La Jolla, CA, USA. ¹¹Max Planck Institute for Medical Research, Heidelberg, Germany. ¹²CeMM Research Center for Molecular Medicine of the Austrian Academy of Sciences, Wien, Austria. ¹³Center for Neuroscience and Cognitive Systems, Istituto Italiano di Tecnologia, Rovereto, Italy. ¹⁴Department of Neurobiology, University of California, San Diego, La Jolla, CA, USA. ¹⁵These authors contributed equally: Abhi Aggarwal, Adrian Negrean, Yang Chen, Rishyashring Iyer. ✉e-mail: hassemanj@janelia.hhmi.org; kaspar.podgorski@alleninstitute.org

## Methods

### Reagent availability

Bacterial (pRSET) and mammalian (pAAV) expression vectors, including different promoters, Cre- and Flp- dependent vectors, and both the NGR and PDGFR membrane targeting sequences, are available from Addgene (IDs: 234435–234454).

### Statistics and reproducibility

Values and errors reported through the text are mean ± s.e.m. unless otherwise noted. Box plots denote quartiles with whiskers extending to minimum and maximum datapoints within $1.5 \times IQR$. All statistical tests are two-sided, unless otherwise noted. Tests that assume normality (for example, t-test) we used only when measurements were confirmed to be normally distributed. Statistical analyses were performed in Graphpad Prism 8, Julia, Python, R and Matlab, as described in the text. Data and code used to produce the figures, and instructions for generating them, are provided in the data supplement.

### Animal care and use statement

All experimental procedures involving animals were performed in accordance with protocols approved by the Institutional Animal Care and Use Committee at the respective institutes (HHMI Janelia Research Campus, Allen Institute, University of California San Diego, TUM, University College London, St. Jude Children's Research Hospital). All procedures performed in the United States conform to the National Institutes of Health (NIH) Guide for the Care and Use of Laboratory Animals. All procedures performed at TUM were approved by the state government of Bavaria, Germany. All procedures performed at UCL conform to the Animals Scientific Procedures Act (1986), and were performed under personal and project licenses released by the Home Office following appropriate ethics review.

> HHMI Janelia Research Campus: protocols 19-176, 22-0214.01
> Allen Institute: protocol 2109
> University of California, San Diego: S02174M
> Technical University of Munich: 2532.Vet_02-23-24 and 2532.Vet_02-21-121
> University College London: animal license PP3929312
> St. Jude Children's Research Hospital: protocol no. 3193
> This research has complied with all applicable ethical regulations.

### Library generation

We first performed random saturating mutagenesis at 41 targeted amino acid residues within iGluSnFR3, screening 96 random variants per site and sequencing only variants that excelled along one or more response variables. The mammalian expression vector pAAV.hSyn. iGluSnFR3.v857.GPI (Addgene no. 178331) and a related variant, iGluSnFR3.v867.GPI, were used as templates for site-directed mutagenesis. This vector contains a BglII restriction site at the N terminus and a PstI restriction site at the C terminus, flanking the iGluSnFR3 gene. To introduce mutations, overlapping internal degenerate-codon primers were designed for each site. Mutagenesis was performed using overlapping paired amplicons, which were assembled into the BglII- and PstI-digested pAAV.hSyn vector using a commercial HiFi DNA Assembly Mix (New England Biolabs). The assembled reaction products were transformed into NEB STABL2 chemically competent *E. coli* cells (New England Biolabs). Transformants were plated on LB agar plates supplemented with ampicillin (100 mg L$^{-1}$) and incubated overnight at 37 °C. For each mutagenesis site, 96 colonies were picked and inoculated into 2.6 ml of 2x-YT medium containing ampicillin (100 mg L$^{-1}$). The cultures were grown for 24 h at 37 °C with shaking at 225 r.p.m. Bacterial cells were then pelleted by centrifugation, and plasmid DNA was isolated using a miniprep kit (Qiagen). The DNA concentrations were normalized to 60 ng µl$^{-1}$, on the basis of absorbance readings at 260 nm using a Tecan Infinite M1000Pro microplate reader. Normalization and dilution of plasmid DNA were automated using a Hamilton Nimbus liquid handling system. Plasmids were electroporated into neuronal cell cultures along with unmutated controls. Variants with the largest positive effect in each plate were selected for Sanger sequencing (Genewiz). Single-site mutations (p.Y31Q, p.Y31E, p.Q34A, p.Q98F, p.A185N, p.T254R, p.K271F, p.K271G, p.H273E, p.Q418S, p.N499L, p.N499V) were selected on the basis of their high performance on at least one of the four response metrics. We generated all possible combinations (1,728) of these 12 mutations. Although the initial saturation screen was performed on iGluSnFR3.v857 and iGluSnFR3.v867 in parallel, owing to poor photostability of iGluSnFR3.v867 derivatives, combinatorial mutagenesis was performed using only pAAV.hSyn.iGluSnFR3.v857.GPI as the template. To generate these combinations, two overlapping amplicons spanning the iGluSnFR3 gene were amplified through PCR. The PCR products were purified and ligated into the pAAV.hSyn vector, which had been digested with BglII and PstI restriction enzymes. The assembled constructs were transformed into NEB STABL2 chemically competent *E. coli* cells (New England Biolabs), resulting in the isolation of 1,728 unique isothermal assembly products in 96-well cultures. These bacterial cultures were grown (225 r.p.m., 37 °C), pelted and miniprepped. The extracted plasmids were confirmed by Sanger sequencing to verify the presence of the intended combinations. Following verification through Sanger sequencing, plasmids were normalized to a concentration of 60 ng µl$^{-1}$ and arrayed for subsequent electroporation into neuronal cell cultures.

A total of 3,365 variants were screened, and 1,640 unique variants were screened in the saturation library (41 sites × 20 residues × 2 templates, assuming full residue coverage by the 96 random samples per variant), 1,728 in the combinatorial library (of which 13 were already present in the saturation screen) and 10 candidate fast variants v8880 (S70A, S70T, S182L, S182V or Y209F) and v8376 (S70A, S70T, S182L, S182V or Y209F).

### Neuron culture screen

Neonatal Sprague–Dawley rat pups, irrespective of sex (Charles River Laboratory), were euthanized, and the hippocampal–cortical cell culture was obtained. Tissue dissociation was carried out using papain (Worthington) in 10 mM HEPES (pH 7.4) prepared in Hanks' Balanced Salt Solution, incubating at 37 °C for 30 min. The resulting cell suspensions were triturated with a Pasteur pipette and filtered through a 40-µm strainer. For transfection, $5 \times 10^5$ viable cells were mixed with 400 ng of plasmid DNA and nucleofection solution in a 25 µL electroporation cuvette (Lonza) and electroporated following the manufacturer's instructions. For the field stimulation assay, neurons were plated at a density of $1 \times 10^5$ cells per well in poly-D-lysine (PDL)-coated, 96-well glass-bottom plates (MatTek, no. 1.5 cover glass). Cells were plated in 100 µl of a medium composed of a 4:1 ratio of NbActiv4 (BrainBits) and plating medium (28 mM glucose, 2.4 mM NaHCO$_3$, 100 µg ml$^{-1}$ transferrin, 25 µg ml$^{-1}$ insulin, 2 mM L-glutamine, 100 U ml$^{-1}$ penicillin, 10 µg ml$^{-1}$ streptomycin and 10% FBS in MEM). On the following day, 190 µl of NbActiv4 medium was added to each well, and plates were incubated at 37 °C with 5% CO$_2$ for 12–15 days before imaging. Typically, 8 wells of a 96-well plate were used to electroporate iGluSnFR3.v857 as a control, while the remaining wells (4 per construct) were used for the constructs of interest. To minimize edge effects, the first and last columns of the plate were not used. Each well was imaged under 1-AP and 20-AP field stimulation[26]. We calculated the following performance metrics using custom software written in Julia v1.11 with LsqFit.jl v0.15.1: brightness, photostability, number of responsive pixels, $\Delta F/F_0$ relative to a 1-s window before stimulation, and rise and decay rates fit to the mean pixel traces. Metrics for each well were normalized to the in-plate iGluSnFR3.v857 controls for analysis.

### Modeling of mutation effects

Analysis of the combinatorial field stimulation screening data was done using a generalized linear model (GLM) approach implemented using

the statsmodels[49] (v0.14.2) package in Python 3.12.2. The input to the GLM is a 9-length categorical prediction vector—there are 9 categories (covering the 9 mutation sites), and each site can have either two or three possible values. The labels provided to the GLM are the values of the sensor performance metrics: $\Delta F/F_0$, $F_0$, $T_{off}$ or $T_{on}$, corresponding to a given variant. For each variant multiple measurements (corresponding to imaging on different days, screens, plates or wells) were used, and no averaging was done. Model parameters for 20 AP field stimulus data is shown in Extended Data Figure 1; 1 AP fits exhibit lower SNR, but were otherwise qualitatively similar. Negative values in $\Delta F/F_0$ were clipped to 0. The appropriate link function and exponential dispersion model (EDM) family was chosen using an exhaustive search, and the ones with the best fit (as assessed by smallest r.m.s.d. on predictions) for each metric were selected. The Poisson family with the log link function was chosen for $F_0$, and $\Delta F/F_0$, and the Gaussian family with the log link function was chosen for $T_{on}$ and $T_{off}$.

Bootstrapping was used to generate confidence intervals of the coefficients in the GLM fits. The fits were repeated by randomly selecting $N$ variants (with repetition) over the dataset, where $N$ is the total size of the dataset. 95% confidence intervals for each coefficient were estimated from the resulting distributions. Coefficients were marked as statistically significant if 0 did not lie within their 95% confidence intervals.

Distances between residues were calculated using the 'get_distance' function from PyMol which calculates the Euclidean distance between the coordinates of two atoms. For the pairwise distance between mutations, this was calculated between the closest atom pair from the residues at those sites.

## Protein crystallization, X-ray diffraction and structure determination
Soluble iGluSnFR3 (pRSET iGluSnFR3.v857, Addgene no. 175186) was subcloned, expressed and purified, as previously described for AspSnFR[50]. Crystallization was performed at 20 °C using the vapor-diffusion method. The protein at a concentration of 18.0 mg ml⁻¹ in 50 mM HEPES, pH 7.3, 50 mM sodium chloride was supplemented with 90 mM L-glutamate before crystallization. Crystals of iGluSnFR3 in complex with L-glutamate were grown by mixing equal volumes of protein solution and precipitant solution containing 26% (mass/vol) PEG 1500. Before flash-cooling in liquid nitrogen, the crystals were briefly washed in a cryoprotectant solution consisting of the reservoir solution supplemented with 20% (vol/vol) glycerol.

Single crystal X-ray diffraction data were collected at 100 K on the ID23-1 beamline at the ESRF (Grenoble, France). All data were processed with XDS[51]. The structure of iGluSnFR3 was determined by molecular replacement (MR) with Phaser[52] using SF-iGluSnFR-S72A coordinates (PDB: 8OVO) as a search model. The final model was optimized in iterative cycles of manual rebuilding using Coot[53] and refinement using Refmac5[54] and phenix.refine[55]. Data collection and refinement statistics are summarized in Supplementary Table 2. Model quality was validated with MolProbity[56] as implemented in PHENIX. Atomic coordinates and structure factors have been deposited in the Protein Data Bank under accession code 9FBU.

## Imaging optical minis
For imaging spontaneous release, $2\times 10^5$ cells were plated onto PDL-coated, 35-mm glass-bottom dishes (Mattek, #0 cover glass) in 120 ml of a 1:1 mixture of NbActiv4 and plating medium in the center of the plate. The next day, 2 ml of NbActiv4 medium was added to each plate. Fifty percent of the medium was replaced with fresh medium at 4 and 7 DIV. Imaging was performed at 14 DIV. Before the experiment, culture medium was replaced with imaging buffer containing the following (in mM): 145 NaCl, 2.5 KCl, 10 glucose, 10 HEPES, pH 7.4, 2 CaCl₂, 1 MgCl₂, 100 mM sucrose (to enhance glutamate release) and 2 mM TTX (to block AP-evoked release). Images were captured with an inverted

fluorescence microscope (Zeiss AXIO Observer 7) equipped with SPECTRA X light engine (Lumencore), a ×63 oil objective (NA = 1.4, Zeiss), and a scientific CMOS camera (Hamamatsu ORCA-Flash 4.0). A FITC filter set (475/50 nm (excitation), 540/50 nm (emission), and a 506LP dichroic mirror (FITC-5050A-000; Semrock)) was used for all iGluSnFR variants in this study. For each neuron culture plate, 6–10 fields of view (FOVs_ were chosen and imaged with constant light intensity at 12 mW mm⁻². For each FOV, 3,000 images (512×512 pixels) were acquired with Hamamatsu image acquisition software (HCImageLive) at 100 Hz.

## Analysis of optical minis
Optical mini recordings were processed using constrained matrix factorization[26]. The SNR for each detected site was calculated as the amplitude of the third largest peak in the 30-second recording, divided by a spectral estimate of the noise amplitude for each trace. The decay time at each site was calculated by detecting isolated peaks in the activity trace, aggregating mean peak-triggered traces, and fitting an exponential decay to the trace following the peak. The median of each statistic across detected sites was calculated for each well, and the mean and SEM of the well medians is shown in Fig. 2d.

The pixelwise statistic used to generate the activity images (red channel in Fig. 2a,e) is the skewness of the highpass filtered image normalized by the square root of the mean intensity. For data derived from a Poisson process (for example, photon shot noise), the expected value of this statistic is 1; values above 1 reflect positive skewness deviations in excess of shot noise.

## Frequency response in hippocampal culture
Sprague Dawley rat hippocampal neurons were isolated at E18 (Envigo) irrespective of sex. Hippocampal tissue was dissected and maintained in chilled hibernate A media (Gibco; 1247501). This tissue was then incubated with 0.25% trypsin (Corning; 25-053 Cl) for 30 min at 37 °C before trituration in DMEM (Gibco; 11965-118) supplemented with 10% fetal bovine serum (FBS, Thermofisher; 501527079) and penicillin-streptomycin (pen/strep, Thermofisher; MT-30-001-Cl). Dissociated neurons were plated on 18 mm glass coverslips (Warner instruments; 64-0734 [CS-18R17]) coated with poly-D-lysine (Thermofisher; ICN10269491). Neurons were initially plated in DMEM with 10% FBS and pen/strep for 2 h at a density of 125,000 cells/well. Then, media was replaced with Neurobasal-A medium (Thermofisher; 10888-022) supplemented with 2% B-27 (Thermofisher; 17504001), 2 mM Glutamax (Thermofisher; 35050031), and pen/strep. On DIV 2, neurons were transfected with 200 ng of plasmid DNA encoding hSyn.iGluSnFR. NGR variants using lipofectamine LTX with PLUS reagent using 1 ml of lipofectamine LTX and 0.5 ml of PLUS reagent per well (Thermofisher; 15338100). Neurons were supplemented with media once a week and imaged at DIV 20.

Imaging was performed on an Olympus IX83 inverted microscope equipped with a X-cite XYLIS LED (Excelitas; XT720S), Olympus 100x/1.5 NA objective (UPLAPO100XOHR), and an ORCA Fusion BT sCMOS camera (Hamamatsu Photonics). Standard imaging media (extracellular fluid; ECF) consisting of 140 mM NaCl, 3 mM KCl, 1.5 mM CaCl2, 1 mM MgCl2, 5.5 mM glucose, 10 mM HEPES (pH 7.3), B27 (Gibco), glutamax (Gibco) was used, with 50 mM D-AP5 (Tocris; 0106), 20 mM CNQX (Tocris; 1045), and 100 mM picrotoxin (Tocris; 1128) added to block postsynaptic signals. Single image planes were acquired with 5.6 ms exposure (178.6 Hz) using Sedat 490/20 nm filter excitation and 525/36 nm filter emission. 400 frames (5.6 ms exposure/ 2240 ms total) were collected, starting 450 ms after the initial frame. A Model 4100 Isolated High Power Stimulator (A-M Systems; 930000) was used with platinum wires attached to a field stimulation chamber (Warner Instruments; RC-49MFSH), integrated by an Axon Digidata 1550B (Molecular Devices). Field stimulus voltage was set to the lowest that reliably produced presynaptic calcium transients in >95% presynaptic boutons using synaptophysin-GCaMP6f as a presynaptic calcium reporter.

Experiments were performed at 33-34 °C. Temperature and humidity were controlled by an Okolab incubation controller and chamber.

## Two-photon imaging screen in mouse cortex

AAV2/1 Viruses encoding iGluSnFR variants in the NGR display vector (a glycosylphosphatidylinositol (GPI) anchor) were generated by the Janelia Viral Core. The NGR vector was chosen based on prior data demonstrating excellent expression and membrane localization in culture[26,37] and preliminary observations that it produced higher expression than the previously-used COBL9 GPI anchor. To better compare performance to a more conventional vector, we included iGluSnFR3 controls of both NGR and PDGFR vectors.

C57BL/6 J male mice (Jackson Laboratory) of 2 months of age were anesthetized and placed in a stereotactic mount. Following craniotomy and dura removal, viral injections were done at two sites in the left visual cortex area V1 (−4.0 A/P, −2.8 M/L, 0.3 D/V and −2.5 A/P, same M/L and D/V coordinates), using a Nanoject III injector and a beveled borosilicate micropipette. Sparse expression necessary to distinguish individual neurites was achieved by injecting 250 nL of high-titer flexed indicator and low-titer Cre AAV mix (vg/mL): 2E12 AAV1-hSyn-Flex-<iGluSnFR > + 1.2E8 AAV9-CamKII-0.4.Cre.SV40 (Addgene 105558-AAV9).

Two-photon microscopy imaging was done on a 12 kHz commercial resonant scanning microscope (Bergamo II, Thorlabs; ScanImage, MBF Bioscience) using an Olympus 25×1.0 NA objective equipped with a spherical aberration correction collar to correct for the imaging window coverslip. Two-photon excitation was done at 1030 nm (Insight X3, Newport) using a fixed 46 mW power post-objective for all iGluSnFR variants and imaging depths. Fluorescence was collected using a GaAsP photomultiplier through a 525/50 band-pass filter (Semrock). For in vivo indicator screening, scans were acquired at 430 Hz, using 50 lines x 128 pixels/line at 10× software zoom, with scans lasting 2 min, while mice were headfixed, awake and visually stimulated with a 1 s on/off diffuse light pulse that illuminated the microscope enclosure. Further functional characterization of select variants was done using moving gratings at different orientations in 45-degree increments.

Custom Matlab scripts were used to motion correct raw scans using a cross-correlation template-maximization approach, followed by automated identification and extraction of iGluSnFR sources and events. In a first pass, the algorithm generated best estimates for source localization on the basis of fluorescence transients, which were clustered into larger patches. These initial source estimates were further refined using non-negative matrix factorization and a least-squares fit to yield source footprints from which fluorescence signals were extracted and further processed to characterize transients.

Several measures were used to compare indicator fluorescence properties: decay time constant, event signal-to-noise ratio (eSNR), event detectability, fluorescence skewness, bleaching factor and baseline fluorescence. On a source level, event decay time constant was calculated by first dividing the recorded fluorescence into 10 s windows, and within each window, the signal was cross-correlated at incremental time-lags up to a maximum lag followed by a robust linear model fit[49] to the logarithm of the cross-correlated time-lags. To obtain the decay time-constant within each 10-s window, only statistically significant linear fits were considered ($P < 0.001$), and was calculated as the absolute value of the inverse of the estimated slope. To reduce bias due to bursts that can obscure single-event fluorescence decay, a branch-level decay time-constant was calculated as the minimum time-constant across all identified sources from the same branch.

Spine turnover measurements were performed on an overlapping cohort of mice as the above screen, using the same surgical preparation. Three mice received AAV1 encoding membrane-tagged GFP (hSyn-FLEX-EGFP.CAAX) in place of iGluSnFR, as a control. Starting 2 weeks after injection, recordings were performed over 3 sessions (days 0, 7, 14) as for the activity screen. Mean motion-registered images from

the recordings were analyzed to quantify spine turnover. Annotations were performed manually through a custom-written Python GUI, blinded to experimental condition.

## In vivo spatial specificity measurements

Two-photon activity imaging was conducted with the same microscope and settings as in the previous section, using an overlapping cohort of mice. To assess spatial specificity of the glutamate indicators, mice were presented with moving grating visual stimuli (2 Hz temporal frequency, 0.05 cycles per degree) of 8 directions (45-degree increments) in randomized order. Stimuli were presented for 2 s following a 1-s mean luminance stimulus. Each grating was presented 15 times during each recording session.

For each video recorded in this way, $\Delta F = F - F_0$ was calculated for each pixel, where $F_0$ is a moving median of $F$ with a window of 10,001 frames. Then, the time-resolved tuning curve was calculated as the average 2-s trace across trials of each orientation presentation. To obtain the eight-point tuning curve, the mean response over the 0.5 s before the stimulus was subtracted from the mean response during the on periods. Preferred orientation was calculated as the direction of the vector sum of response vectors, where each vector's direction is the corresponding stimulus orientation, and the vector magnitude is the response magnitude from the eight-point tuning curve. The orientation selectivity index (OSI) was calculated as the magnitude of the vector sum of response vectors. The response amplitude was calculated as the average of the magnitudes of response vectors. Pixelwise tuning maps were drawn using the preferred orientation as the hue, OSI as the saturation and response amplitude as the value for each pixel.

To quantify degree of spatial specificity, cross-validated covariances were calculated for pairs of pixels. For each pair of pixels, the time-resolved tuning curve was calculated for each pixel using a mutually exclusive random split of trials, and the covariance between tuning curves was calculated. This was done over 20 random splits for each pair of pixels, and the resulting cross-validated covariance was the average of covariances over the 20 splits. This analysis was done separately on the set of labeled pixels (pixels on or near the brightest pixels) and background pixels (all other pixels).

The distance it took for the average cross-validated correlation across recordings to reach $1/e$ and $1/e^2$ of the maximum were recorded. Confidence intervals for these statistics were calculated by bootstrapping on recordings and using 2.5th and 97.5th percentile of the bootstrap distribution from 1,000 bootstrap samples. The difference in statistic was deemed significantly different if the 95% bootstrap confidence interval of the difference in statistic did not include 0.

## In vivo AO-2P imaging and data analysis

WT mice (C57BL/6 J, male, 6–8 weeks old, Jackson Laboratory) were anesthetized with isoflurane using a precision vaporizer (3% vol/vol for induction and 1–2% vol/vol for maintenance) along with subcutaneous injection of buprenorphine (0.1 µg per body weight). Body temperature was maintained at 37 °C with a heating pad during surgery. The animal was placed in a stereotaxic frame. A small hole (~0.2–0.5 mm) was drilled at 1.7 mm posterior to the bregma and 1.8 mm lateral from the midline, corresponding to the transverse location of the barreloids of the right VPM thalamic nucleus. Cre-dependent iGluSnFR3.v857.GPI, iGluSnFR4s.PDGFR and iGluSnFR4f.PDGFR were diluted and mixed to the final titer of $8 \times 10^{12}$ to $1.1 \times 10^{13}$ GC ml$^{-1}$ for the GluSnFR virus and 3 $\times 10^{10}$ for the Cre virus. The viruses were injected into VPM thalamus at a depth of 3.25 mm and 3.1 mm at 1.6 mm posterior, 1.8 mm lateral and 1.65 mm posterior, 2.0 mm lateral at 50 nl, 10 nl min$^{-1}$ at each location. A 4-mm craniotomy was performed over the right vS1 cortex (centroid at 1.6 mm posterior to the bregma and 3.3 mm lateral from the midline). A cranial window by a single 4-mm round coverslip (no. 1) was embedded in the craniotomy and sealed with cyanoacrylate glue (Loctite, catalog no. 401). Meta-bond (Parkell) was further applied around the edge to

reinforce stability. A titanium head-bar was attached to the skull with Meta-bond, and the remaining exposed bone was covered with dental acrylic (Lang Dental).

In vivo imaging was carried out after 4 weeks of expression and 3 days of habituation for head fixation. All imaging experiments used head-fixed awake mice under AO-2P. AO correction was applied at imaging depths below 350 μm[57]. For functional imaging, the laser was tuned to 960 nm, with a post-objective power of 80–100 mW. The frame rate was 240 Hz. The vibrissae of the mice were trimmed 3 days before functional experiments, leaving only two vibrissae among C1, C2, D1, D2 or Beta, whose corresponding cortical column had an optimal expression of iGluSnFR. The mice were head fixed to the imaging rig, together with a running disk. The vibrissa was illuminated with an infrared LED (M940L3, Thorlabs) and captured with a camera operated at 500 Hz and synchronized to the imaging setup.

Air-puff deflection was used for vibrissa stimulation. Pulse-controlled compressed air, 20-ms pulse width, 5 p.s.i. at the source, was delivered through a fine tube, which was placed parallel to the side of the mouse snout and 20 mm away from the targeted vibrissa. The frequency of the air puffs was from 2 to 30 Hz. The experiments were carried out in the absence of visible light. Within each barrel, eight regions (30×60 μm) were imaged at each stimulus frequency in random order to avoid any bias from photobleaching.

The frames in the raw two photon recordings were motion corrected[58] and background subtracted. The average and standard deviation of the frames were sent to a bouton mask extraction algorithm using adaptive thresholding and estimating the connected components that were within the expected size for individual boutons. The fluorescence signals of each bouton were linearly interpolated. Free-whisking experiments were conducted by placing a thin pole on a stage capable of moving randomly in the rostro-caudal and medio-lateral directions, where the pole is occasionally within the reach of the mouse vibrissae. The data were recorded while the mouse was in its exploratory phase, during which the vibrissae swept rhythmically. Vibrissa location, and structure along with the touch events, were identified using a maskless posture estimation package[59].

## In vivo two-photon imaging and cell-attached recordings

Adult C57Bl/6 (8–16 weeks) male mice were anesthetized with isoflurane (2% for induction and 1.5% for maintenance, in pure oxygen). The body temperature was continuously monitored and kept at 37.5 °C by a heating plate. Both eyes were covered with ophthalmic ointment. The skin above the left hemisphere was carefully removed by fine scissors to expose the skull following the administration of an analgesic (Metamizole, 200 mg kg⁻¹ body weight) and a local anesthetic (2% xylocaine). The exposed skull was left dry and cleaned with a sterile scalpel blade. A custom-made headplate was fixed to the cleaned skull with light-cured dental cement. For the chronic window, a 3-mm circular craniotomy was made above the left V1, and a matching coverslip was glued to the craniotomy using vetbond. For single-cell plasmid electroporation, the coverslip on the craniotomy had a small perforation that allowed the access of a patch–pipette to the cortical tissue. The procedure for single-cell electroporation was the same as previously described[26]. In brief, plasmids encoding pAAV.hSyn.iGluSnFR variants in the PDGFR backbone were dissolved in an artificial intracellular solution to a concentration of ~100 ng μl⁻¹ and was delivered to random layer 2/3 neurons using a 4–5 MΩ patch–pipette. After electroporation, the perforated coverslip was carefully removed and replaced by an intact one. The hippocampal window was similarly prepared. After a 3.0-mm craniotomy was made on the right hemisphere, the cortical tissue was carefully removed with a blunt needle that was connected to a vacuum pump. The corpus callosum and external capsule were also removed to expose the alveus of the hippocampus. The hippocampal window assembly (consisting of a coverslip and a stainless-steel cannula) was implanted so that the coverslip was directly on top of the hippocampus.

Two-photon glutamate imaging was performed with a custom-built two-photon microscope based on a commercial upright microscope (Slicescope, Scientifica). The microscope featured a 12-kHz resonant scanner (Cambridge Technology) and a tunable Ti:sapphire laser (Mai Tai HP DeepSee, Spectra-Physics). Microscope control and data acquisition were achieved using a custom LabVIEW GUI that supported a frame rate of up to 500 Hz. The laser was tuned to 950 nm for glutamate imaging and the power under the objective (×40, NA 0.8, Nikon) was kept below 30 mW to prevent photodamage.

Dendrites of layer 2/3 pyramidal neurons in mouse V1 were imaged at 200 Hz frame rate and glutamate signals from individual synapses were probed using full-field square wave drifting gratings (0.03 cycle per degree spatial frequency, 2 Hz temporal frequency, 2 s duration, 8 directions). Drifting gratings stimuli were generated using a custom Android app and presented on a tablet (Samsung, 10.5-inch) equipped with an OLED display. The tablet was placed 10 cm in front of the right eye of the mouse, covering a visual space of 96 degrees in azimuth and 70 degrees in elevation, and had a mean brightness of 4.8 cd cm⁻².

For simultaneous two-photon axonal imaging and cell-attached recordings, the coverslip was changed to one with an access opening and warm artificial cerebro-spinal fluid (ACSF) (125 mM NaCl, 26 mM NaHCO₃, 4.5 mM KCl, 2 mM CaCl₂, 1.25 mM NaH₂PO₄, 1 mM MgCl₂ and 20 mM glucose) was perfused throughout the experiment to keep the brain temperature constant. Loose seal cell-attached recordings from glutamate-sensor-expressing cells were performed with a patch-pipette filled with ACSF containing 25 μM Alexa Fluor 594. Axonal boutons from the same neuron were identified and imaged at a frame rate of 500 Hz. Electrophysiological signals were acquired in either current–clamp mode or voltage–clamp mode by a patch–clamp amplifier (EPC10, HEKA).

All data were processed using custom scripts in MATLAB (v2024b), and statistical tests were conducted in R (v4.4.2). ROIs were automatically segmented or manually drawn on the basis of the local correlation image. The mean fluorescence signal ($F$) from each ROI was extracted, and the fluorescence change was calculated as $\Delta F/F = (F - F_0)/F_0$. $F_0$ was the mean of baseline fluorescence. All traces were denoised using a first-order Savitzky–Golay filter (25 ms window) except for the kinetics analysis.

To quantify the kinetics of each variant, spike-triggered averages of single-AP-associated glutamate signals were calculated for each axonal bouton. Because of the much slower time course of iGluSnFR4s, only single APs separated by more than 500 ms from the preceding and following APs were selected. The resulting single-AP glutamate transients were fitted to the following exponential equation.

$$\Delta F/F(t) = A\left(1 - e^{-(t-t_0)/\tau_{\text{rise}}}\right) e^{-(t-t_0)/\tau_{\text{decay}}}$$

where $A$ is the amplitude factor, $t_0$ is the time of the AP and $\tau_{\text{rise}}$ and $\tau_{\text{decay}}$ are time constants for rise and decay, respectively. The amplitude was measured as the peak of the fitted trace, and time-to-peak was the time between the spike and the peak.

To compare the two iGluSnFR4 variants in resolving synaptic orientation selectivity in the visual cortex, the orientation tuning curve for individual synapses was obtained by calculating either the amplitude or the mean of the response. The tuning curve was then mapped onto orientation vector space and summed. The preferred orientation was the angle of the resultant vector. Orientation selectivity index (OSI) was calculated as the length of the resultant vector divided by the sum of responses to each orientation. The tuning magnitude was defined as the 2-norm of the tuning curve. For Pixelwise orientation maps, the hue is the preferred orientation (normalized to 180 °), the saturation corresponds to OSI, and the lightness is the tuning magnitude (normalized to the maximum of each field of view). To compare recordings from the hippocampus with data from V1, the denoised $\Delta F/F$ trace was converted to a modified $z$-score trace using the mean and the s.d. from

a baseline period. The baseline period is a 0.5-s period with the lowest s.d. Spontaneous glutamate transients were automatically detected using a threshold of eight times the baseline standard deviation.

## Retinotopic mapping of L5 neuron dendrites

Experiments were performed on six wild-type mice (C57Bl6/J), aged between 4 and 6 weeks, of both sexes. Animals were anaesthetised with isoflurane (1–2% in Oxygen), their body temperature was monitored and kept at 37–38 °C using a closed-loop heating pad, and the eyes were protected with ophthalmic gel (Viscotears Liquid Gel, Alcon). Analgesic (Rimadyl, 5 mg per kg body weight) was administered subcutaneously before the procedure, and orally on subsequent days. Dexamethasone (0.5 mg per kg body weight) was administered intramuscularly 30 min prior to the procedure to prevent brain edema. The surgery began with the implant of a head-plate over the right hemisphere of the cranium. The head was shaved and disinfected; the cranium was exposed and covered with biocompatible cyanoacrylate glue (Vetbond, 3 M). A stainless-steel head plate with a 10-mm circular opening was secured over the skull using dental cement (Super-Bond C&B, 10 Sun Medical). Then a 3-mm-wide square craniotomy was opened over V1 (centred at −3.3 mm AP, 2.8 ML from bregma). The exposed brain was constantly perfused with artificial cerebrospinal fluid (150 mM NaCl, 2.5 mM KCl, 10 mM HEPES, 2 mM CaCl₂, 1 mM MgCl₂; pH 7.3 adjusted with NaOH, 300 mOsm). We delivered 60-nl injections at multiple sites, at a cortical depth of ~500 µm, to virally express iGluSnFR variants in layer 5 pyramidal neurons. We achieved sparse expression by injecting diluted AAV1-CaMK2a-Cre ($2 \times 10^8$ GC per ml), mixed with a concentrated: AAV1 hSyn.FLEX.iGluSnFR3.v857.GPI (final concentration: $3.88 \times 10^{12}$ GC per ml; 3 mice); or pGP-AAV-hSyn.FLEX.iGluSnFR4.v8880.PDGFR (final concentration: $2.3 \times 10^{12}$ GC per ml; 2 mice). For other purposes, the injection mix included also AAV1.hSyn.FLEX.NES-jRGECO1a.WPRE. SV40 (final concentration: $2.2 \times 10^{12}$ GC per ml). Following the injections, the craniotomy was sealed with a glass cranial window, assembled from a circular cover glass (4 mm diameter, 100 µm thickness) glued to a smaller insert (3 mm wide, 300 µm thickness) with index-matched UV curing adhesive (Norland no. 61).

Recordings of neuronal activity were performed 1 month after surgery with a standard resonant-scanning two-photon microscope (Bergamo II, Thorlabs), equipped with a Nikon ×16, 0.8 NA objective. The microscope was controlled using ScanImage 2023.1 (MBF Biosciences). Excitation light was provided by a femtosecond laser (Chameleon Discovery TPC, Coherent) at 950 nm. Laser power was depth-adjusted between 25 and 50 mW. Sample fluorescence was collected in the green (525/50 nm) band. During recordings, mice were head fixed over a metal mesh wheel, where they could run at will. For each FOV, a structural z-stack served as a reference to identify L5 neurons and trace their apical dendrites back to the surface. For each neuron, as many apical dendrites as possible were recorded in sequential and longitudinal acquisitions, at a cortical depth ranging from 10 µm to 100 µm below the brain surface. Functional imaging was performed over fields of view of 512 × 512 pixels at 30 Hz, with a resolution ranging from 0.1894 to 0.1088 µm per pixel.

Visual stimuli were generated in Matlab (MathWorks 2021b) and displayed on three gamma-corrected LCD monitors (Adafruit 1.8" TFT LCD Display, resolution 128×160 px) surrounding the mouse at 90 degrees to each other. The LCD screens were covered with Fresnel lenses to correct for viewing angle inhomogeneity of the LCD luminance. The mouse was positioned at the center of the U-shaped monitor arrangement at 20 cm from all three monitors, so that the monitors spanned ±135 degrees of horizontal and ±35 degrees of the vertical visual field. Sparse, spatial white noise stimuli were used to map the retinotopy of the imaged area and estimate the receptive field (RF) of spine inputs. Patterns of sparse black and white squares (4.5–6 degrees of the visual field) on a gray background were presented at 5 Hz, typically in 10-min sequences repeated 3 times for each FOV. At any point in time, each square had a 2% probability of being non-gray, independent of the other squares.

## Analysis of retinotopic mapping data

Recordings were pre-processed using Suite2p[44] (v0.14.0). The final selection of ROIs was curated manually to eliminate noisy ROIs and retain only dendritic branches from the target neuron in a given imaging session. All subsequent analyses were performed in MATLAB R2021b (MathWorks). Linear neuronal spatiotemporal RFs were estimated by regularized linear regression between the neuronal responses and the stimulus history. The Laplacian of the receptive field (RF) in space was regularized to enforce smoothness. stRFs were modeled using time-lagged regression: the stimulus matrix was constructed as the Toeplitz matrix containing the stimulus history shifted at different lags between $t = 0$ and $t = 1.4$ s. ON and OFF subfields were fit simultaneously using separate stimulus predictors for increases and decreases in luminance at the same pixel, respectively. The ON/OFF subfield is computed as the average of the ON and OFF subfields while considering their respective signs to maintain the correct polarity.

For each ROI, we used threefold cross-validation to choose the regularization parameter that maximized the stRF predictive performance. The RF performance was measured as the explained variance (EV), defined as:

$$EV = 1 - \sum (y_{\text{test}} - \hat{y})^2 / \sum (y_{\text{test}} - \bar{\hat{y}})^2$$

where $y_{\text{test}}$ is the actual neuronal response, $\hat{y}$ is the predicted response and $\bar{y}$ is the mean response. The final stRF was then fit with the best $\lambda$ fixed, over the full recording.

We estimated stRF significance using a circular shift test. For each ROI, we fit the RF after circularly shifting the stimulus predictor in time, and repeated this procedure 1,000 times to build a null distribution of explained variances. A P value was then calculated as the fraction of the explained variance from the null distribution that exceeded the explained variance from the real, unshifted data. ROIs were considered responsive if they met two criteria, (1) explained variance from the original dataset exceeded 0.1, and (2) the P value from the circular shift test was smaller than 0.005.

The spatial RFs of responsive ROIs were estimated as the slice of the stRF bearing the largest response and were subsequently fit with a 2D Gaussian. This Gaussian fit represents a localized elliptical region where the ROI exhibits a response.

## Fiber photometry

Surgeries for in vivo fiber photometry recordings (Fig. 5) were performed according to our published protocol[60]. In brief, we performed bilateral (100–150 nl each hemisphere, $2.0 \times 10^{12}$ GC ml⁻¹) injections of AAV (php.eB) encoding hSyn.FLEX.iGluSnFR variants into the VTA of Gad2-ires-Cre mice (JAX 028867). In each mouse, one random hemisphere received iGluSnFR4.PDGFR and the other either iGluSnFR3.v857.PDGFR or SF-iGluSnFR.A184S.PDGFR. These paired measurements accounted for systematic artifacts across pairs, owing to their similar pH and aspartate sensitivities. We recommend the use of yGluSnFR-Null[42] or AspSnFR[50] as glutamate-insensitive controls for iGluSnFR4 photometry. The coordinates targeting VTA were (AP: −3.05; ML: ± 1.75; DV: −4.2 from pia; ±15 degree). After a minimum two-week recovery period following surgery, the mice underwent mild water restriction. We used a custom made CMOS-based fiber-photometry system according to our published protocol[61]. In brief, both GFP-based SF-iGluSnFR and YFP-based SF-iGluSnFR were excited with a 470 nm LED and emission signals were collected with a conventional GFP emission filter (520 ± 35 nm). CMOS sensors and LEDs were controlled by a Teensy 4.1 microcontroller and images from CMOS sensors were acquired using a custom program written in Bonsai[62]. The same Bonsai instance was used to generate sequences of paired stimuli consisting

of a tone (5 kHz pure-tone, ~70 dB, 1 s duration) followed by a 1 s delay and a subsequent water reward (2 µL). Inter-trial intervals were randomized. To specifically examine reward responses independent of learned associations, we analyzed only the initial sessions before mice had developed associations between the tone and reward delivery.

Data were preprocessed and analyzed using custom Python scripts. The time course of raw CMOS pixel values was detrended using a mild Gaussian low-cut filter (cutoff: 2–4 min) and corrected for slow baseline drift caused by photobleaching using a 4th-order polynomial fit. For generating peri-event time histograms (PETH; Fig. 5c), traces from individual trials were locally baseline-subtracted using the 2-s period preceding reward consumption, and then averaged across trials.

### Reporting summary
Further information on research design is available in the Nature Portfolio Reporting Summary linked to this article.

### Data availability
Data used to produce the figures are included in the code and data supplement (https://doi.org/10.25378/janelia.30251743).

### Code availability
Custom code used to produce the figures are included in the code and data supplement under the CC BY 4.0 license (https://doi.org/10.25378/janelia.30251743).

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

### Acknowledgements

We thank the Lab Animal Services and Neurosurgery & Behavior teams at the Allen Institute for technical support, B. Wynalda and S. Narayan for assistance in cloning and M. Seyedolmohadesin for assistance with long-duration in vivo imaging experiments. We thank T. Brown, the Tool Translation Team and D. Walpita (Janelia Research Campus) for providing resources and access to instrumentation. We thank I. Schlichting for X-ray data collection, the European Synchrotron Radiation Facility (ESRF) for provision of synchrotron radiation facilities and E. Mossou for assistance and support in using beamline ID23-1. Work at the Allen Institute was funded by grants from the National Institutes of Health (NIH), specifically, 1DP2NS136990 and BRAIN Initiative UM1MH136462 to K.P. and 1F30MH138009 to M.E.X., by the Paul and Daisy Soros Fellowships for New Americans to M.E.X., by the Human Frontier Science Program HFSP(LT0052/2022-L) to K.M.H. and by the Allen Institute. Work at Janelia Research Campus was funded by the Howard Hughes Medical Institute. Work at TUM was funded by the Deutsche Forschungsgemeinschaft (KO 979/7-1) and the Max Planck School of Cognition. A.K. is a Hertie-Senior-Professor for Neuroscience. Work at UCL and IIT was funded by the Armenise-Harvard Foundation (CDA to L.F.R.), the Human Technopole (HT–ECF 3588 to L.F.R.), the Boehringer Ingelheim Foundation (studentship to A.L.), the Biotechnology and Biological Sciences Research Council (studentship to A.L.) and by UKRI (Frontier Award EP/X022366/1 to M.C.). Work at St. Jude Children's Research Hospital was funded by the American Lebanese Syrian Associated Charities (ALSAC). Work at UCSD was funded by grants from the National Institutes of Health (NIH), specifically, U19 NS137920, U24 EB028942, R01 NS143141 and U01 NS126054 to D.K. R.I. is a Schmidt Science Fellow.

### Author contributions

Conceptualization: A.A., J.H. and K.P. Data curation: A.A., A.N., Y.C., R.I., D.R., A.L., M.E.X., K.M.H., L.W.K., J.L.S., P.Y., B.J.A. and K.P. Formal analysis: A.A., A.N., Y.C., R.I., D.R., A.L., A.P., M.E.X., K.M.H., L.W.K., J.L.S., J.Z., A.T., B.J.A., M.T., S.C.T., A.G.T., L.F.R., D.K. and K.P. Funding acquisition: A.L., M.C., M.E.X., L.F.R., D.K., A.K., K.S. and K.P. Investigation: A.A., A.N., Y.C., R.I., D.R., A.L., M.E.X., B.J.M., K.M.H., L.W.K., J.L.S., P.Y., J.Z., A.T., G.T., R.H.P., J.H., P.L., M.T., J.S.M., L.F.R., J.H. and K.P. Methodology: A.A., A.N., Y.C., R.I., D.R., M.E.X., K.M.H., J.Z., A.T., G.T., J.H., P.L., M.T., S.C.T., A.G.T., M.C., L.F.R., D.K. and K.P. Project administration: A.A., J.H., S.C.T., A.G.T., D.K., A.K., G.C.T., J.H. and K.P. Resources: D.R., J.H., M.T., J.D.V., S.C.T., A.G.T., M.C., D.K., K.S., A.K., J.H. and K.P. Software: Y.C., D.R., M.E.X., K.M.H., L.W.K., B.J.A. and K.P. Supervision: A.A., J.H., J.D.V., A.G.T., M.C., L.F.R., D.K., A.K., K.S., G.C.T., J.H. and K.P. Validation: A.A., A.N., Y.C., R.I., D.R., M.E.X., K.M.H. and K.P. Visualization: A.A., A.N., Y.C., R.I., D.R., A.P., M.E.X., K.M.H., L.W.K., J.L.S., L.F.R. and K.P. Writing—original draft: A.A., A.N., Y.C., R.I., A.L., M.E.X., M.T., S.C.T., A.G.T., L.F.R., D.K., A.K., G.C.T. and K.P. Writing—review and editing: A.A., K.S., G.C.T., J.H. and K.P.

### Competing interests

The authors have no competing interests related to this work.
The funders had no role in study design, data collection and analysis, decision to publish or preparation of the manuscript.

### Additional information
**Extended data** is available for this paper at https://doi.org/10.1038/s41592-025-02965-z.

**Correspondence and requests for materials** should be addressed to Jeremy P. Hasseman or Kaspar Podgorski.

**Peer review information** *Nature Methods* thanks Yi Zuo, and the other, anonymous, reviewer(s) for their contribution to the peer review of

this work. Peer reviewer reports are available. Primary Handling Editor: Nina Vogt, in collaboration with the *Nature Methods* team.

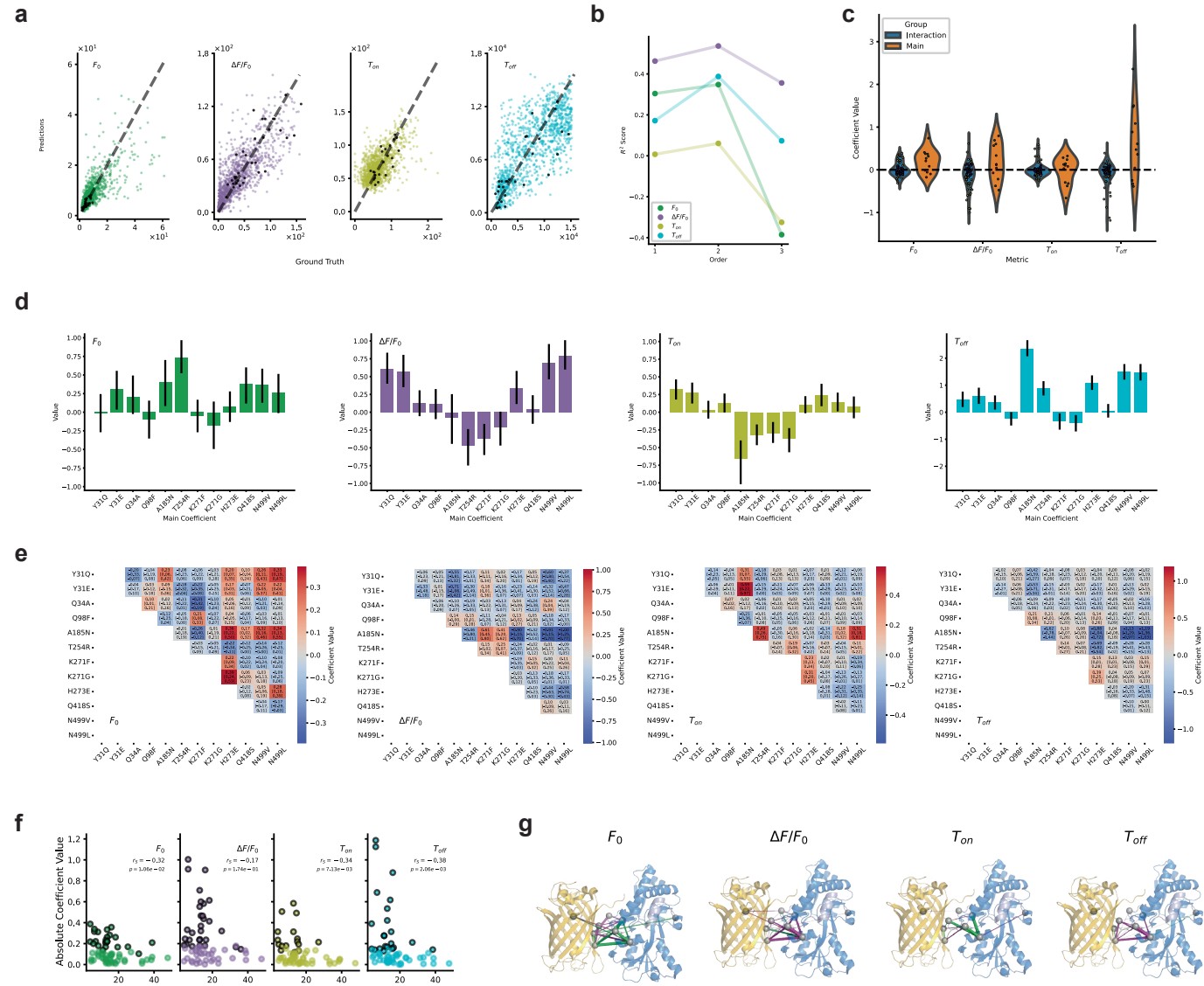

**Extended Data Fig. 1 | Analysis of mutation effects and spatial interactions.**
**a)** Measured variant mean values and corresponding predictions of a second-order GLM for each of the four response variables F0,(baseline), ΔF/F₀ (fractional response), T_on (rise time constant), and T_off (decay time constant) (with the largest 1% of outliers clipped). The points in black correspond to variants that contain only a single mutation from the 12 mutations in the combinatorial screen.
**b)** Cross-validated variance explained for GLMs of different orders: 1 (main effects only), 2 (main effects and all pairwise interactions), 3 (main effect and all pair and triplet interactions). The order 2 model was used in this study. The yellow (in main) and blue (in interaction) points represent statistically significant coefficients (p < 0.05, bootstrap test). **c)** Distributions of coefficients for main effects (orange) and interactions (blue) for the response variables. 34 of 48 main effects and 92 of 252 interactions were individually statistically significant at 95% confidence (bootstrap test). **d)** Main effects of each mutation and bootstrapped 95% confidence intervals. **e)** Interaction terms for each mutation

pair and bootstrapped 95% confidence intervals. **f)** Pairwise distances and coefficient magnitudes for interaction terms, and corresponding Spearman's correlation coefficients. The outlined points represent statistically significant coefficients (p < 0.05, bootstrap test). **g)** iGluSnFR3 glutamate-bound crystal structure solved in this work (PDB: 9FBU), with significant interactions overlaid (p < 0.05, bootstrap test) for each response variable. Green links denote positive interactions. Purple links denote negative interactions. Line thickness proportional to coefficient magnitude. On average across all metrics, the number of statistically significant interactions (p < 0.05, bootstrap test) between residues are of the following types: 3%/8% FP-FP, 8%/32% FP-BP, 6%/14% BP-BP, 12%/24% BP-Linker, and 1%/3% Linker-Linker (where the numerator is the percentage of statistically significant interactions of the total number of interaction coefficients, and denominator is the percentage of possible interactions of the given interaction type of the total interactions).

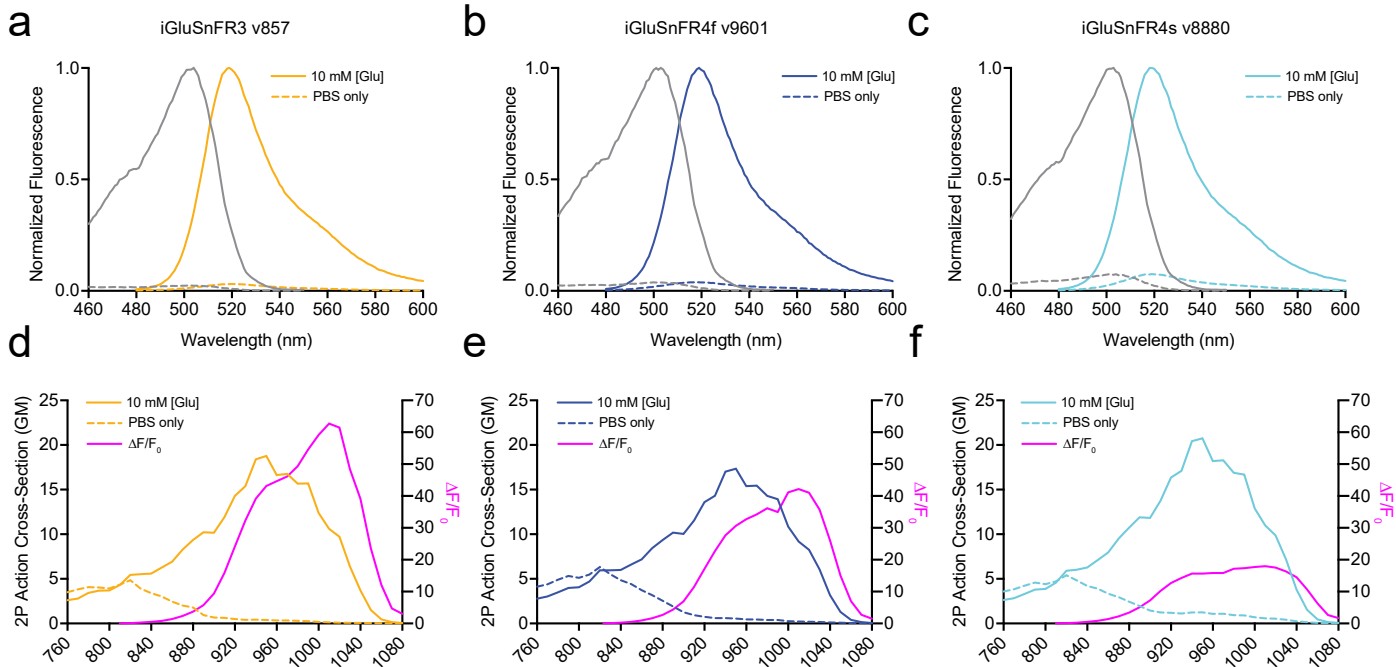

**Extended Data Fig. 2 | 1 P and 2 P spectra of iGluSnFR variants. a-c)** One-photon excitation and emission spectra of iGluSnFR3 v857, iGluSnFR4f, and iGluSnFR4 in the presence (10 mM) and absence of glutamate. **d-f)** Two-photon spectra and ΔF/F₀ of the iGluSnFR4 variants in the presence (10 mM) and absence of glutamate.

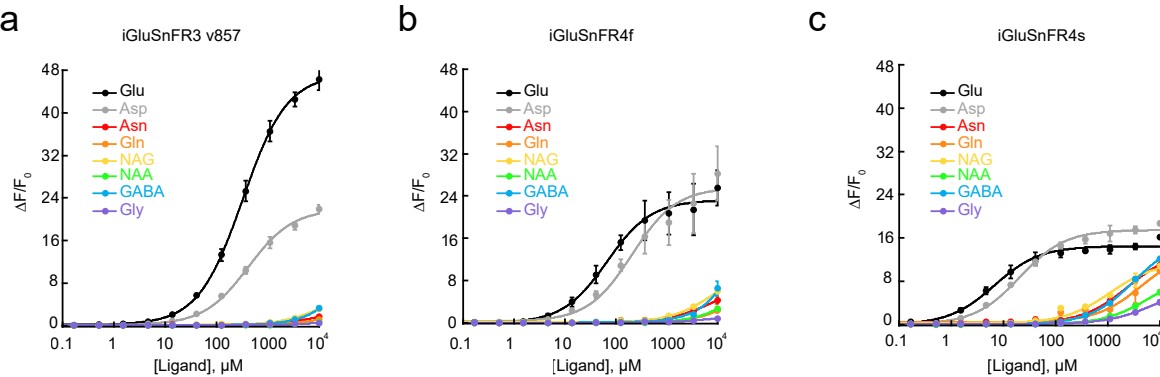

**Extended Data Fig. 3 | Ligand specificity of purified soluble proteins.** $\Delta F/F_0$ of iGluSnFR3 **(a)**, iGluSnFR4f **(b)** and iGluSnFR4s **(c)** for titrations of selected L-amino acids, neurotransmitters, and other drugs (pH 7.3, buffered in PBS). All measurements were made using purified soluble protein; N = 3 titration series of a single protein sample for each measurement.

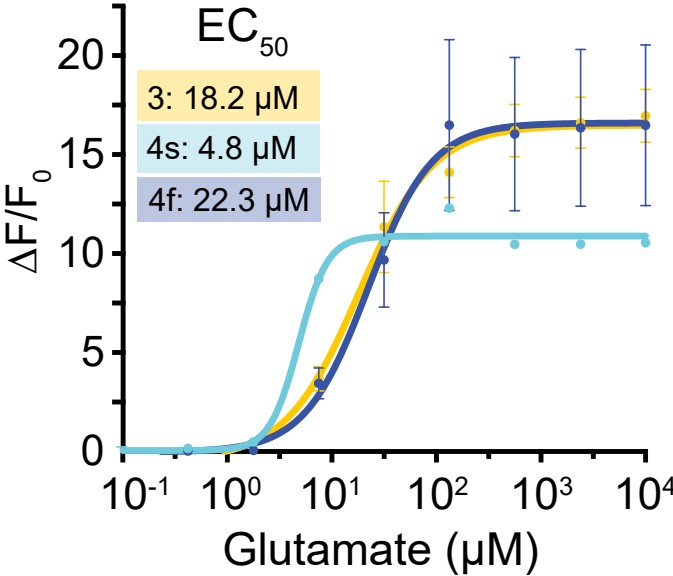

**Extended Data Fig. 4 | Glutamate titrations on the surface of cultured neurons.** Titration curves for iGluSnFR4 variants expressed on the surface of cultured neurons. Data points represent the mean of N = 3 replicates, with error bars indicating SEM. Sigmoidal fits overlaid.

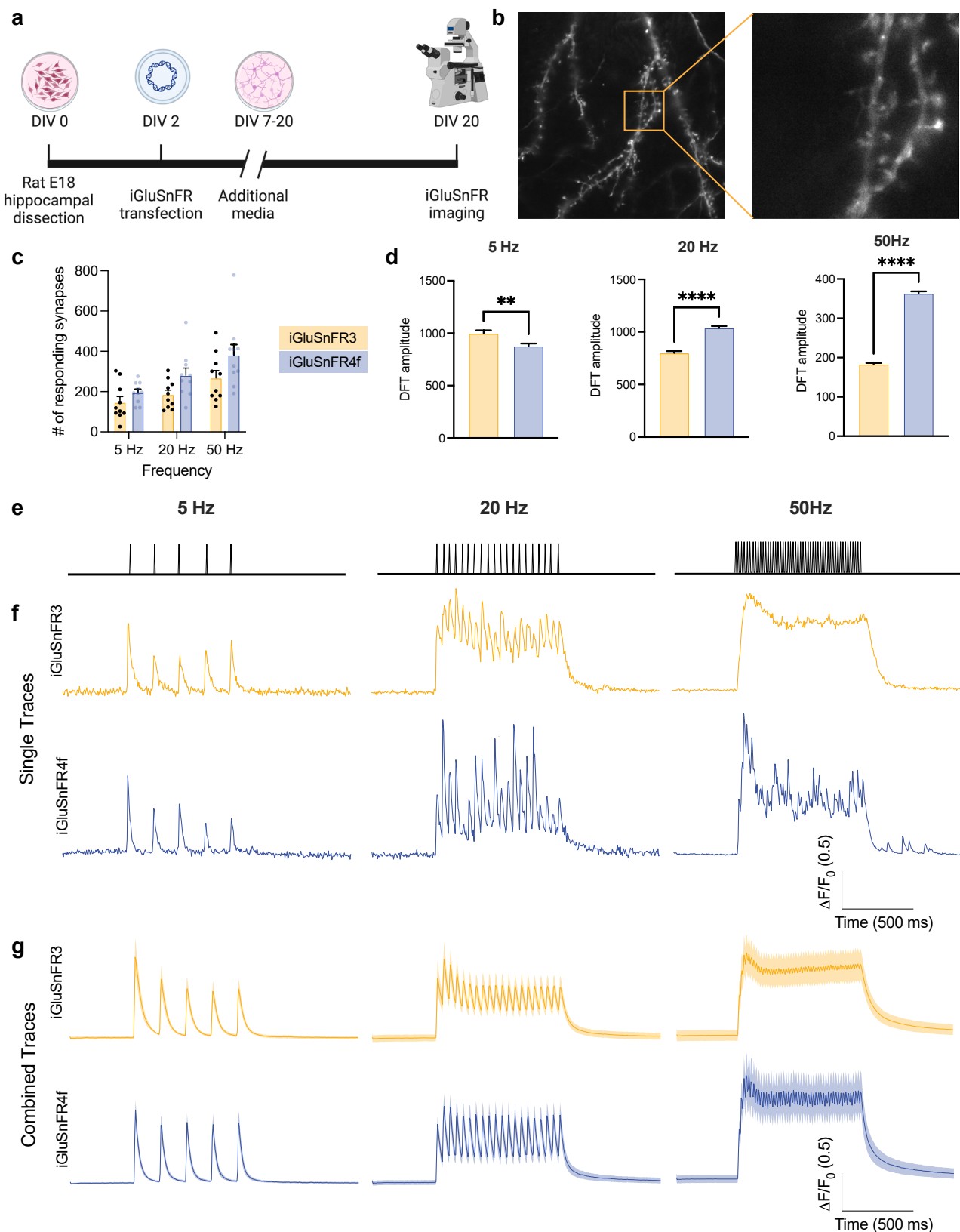

**Extended Data Fig. 5 | See next page for caption.**

**Extended Data Fig. 5 | High frequency stimulation in hippocampal neuron culture. a)** Experiment scheme. Hippocampi from E18 rat pups were dissected (DIV 0) and transfected with iGluSnFR plasmids at DIV 2. Neurons were given additional media from DIV 7-20, then iGluSnFR imaging was performed on DIV 20. Made with Biorender. **b)** Representative field of view of a dendritic arbor expressing iGluSnFR. **c)** Number of synapses responding to either 5 Hz, 20 Hz, or 50 Hz stimuli for iGluSnFR3 and iGluSnFR4f. n = 10 neurons for each condition. Data shown is the mean ± standard error of the mean. **d)** Power spectral amplitude of evoked responses at 5 Hz, 20 Hz and 50 Hz. Bar graphs represent the mean ± standard error for n = 10 neurons per variant. Unpaired t-test 5 Hz p = 0.0025; 20 Hz p = 3.41081E-19; 50 Hz p = 2.359E-157). **e)** Representative stimulation paradigms for 5 Hz/5 stimuli, 20 Hz/20 stimuli, and 50 Hz/50 stimuli. **f)** Traces from single synapses for both iGluSnFR3 and iGluSnFR4f in response to either 5 Hz, 20 Hz or 50 Hz stimuli. **g)** Combined traces from n = 10 neurons for iGluSnFR3 and iGluSnFR4f in response to either 5 Hz, 20 Hz or 50 Hz stimuli. Data shown is the mean ± standard deviation.

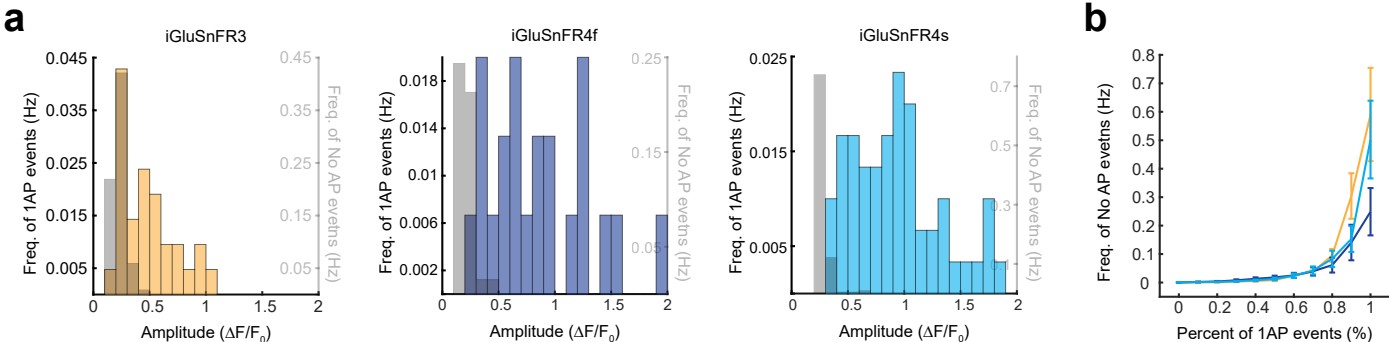

**Extended Data Fig. 6 | Detection of single action potentials by iGluSnFR variants. a)** Amplitude histograms for single-AP events in V1 axons for iGluSnFR3, 4 f, and 4 s, and events without associated APs at the corresponding amplitude thresholds in grey. **b)** Frequency of false positives (no AP events) as a function of fraction of true positive 1 AP events detected, for iGluSnFR3 (yellow), 4 f (navy blue) and 4 s (cyan).

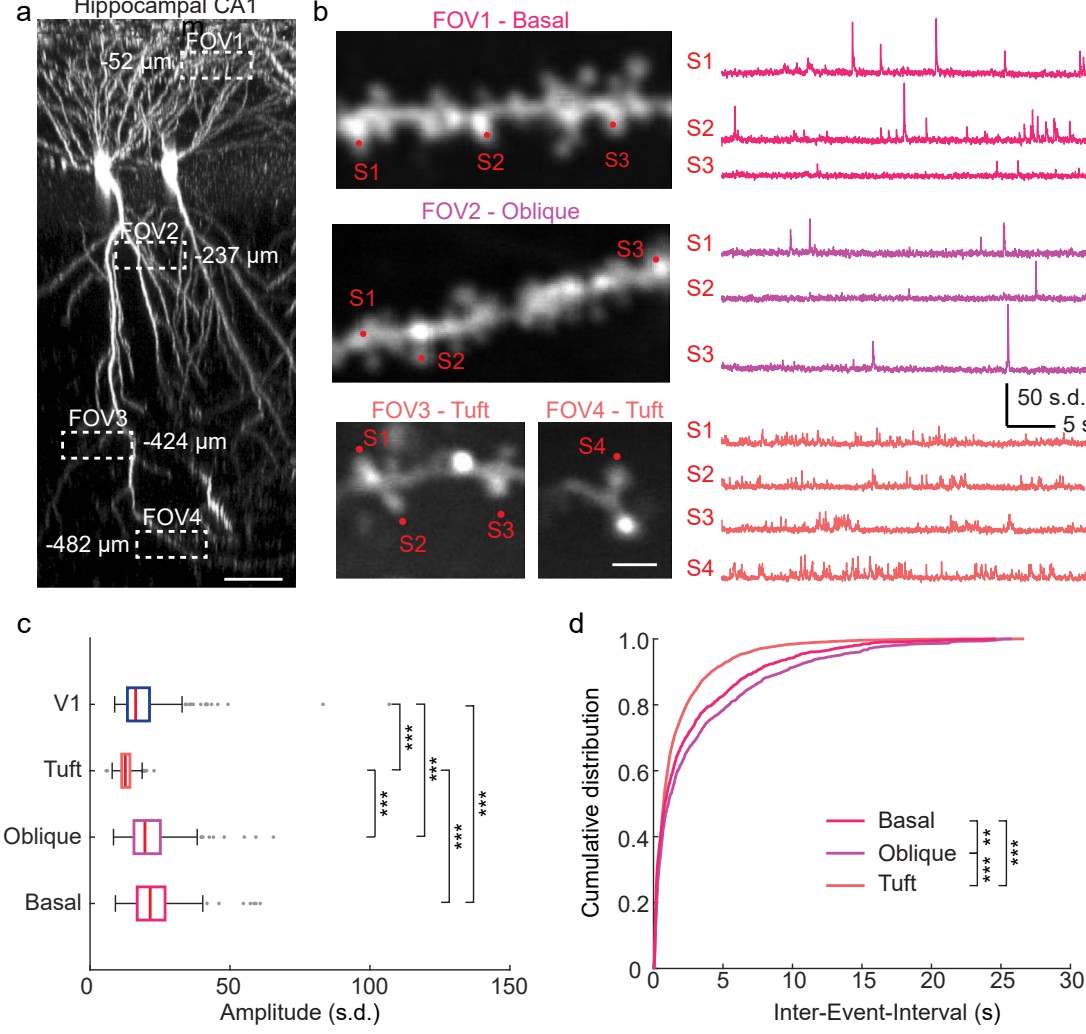

**Extended Data Fig. 7 | iGluSnFR4f imaging in hippocampus CA1.**
**a)** Z-projection of iGluSnFR4f labeled CA1 pyramidal neurons and locations of imaged dendritic segments. The number next to the box indicates the depth below the coverslip. Scale bar, 50 μm. **b)** Two-photon images of dendritic segments from basal, oblique, and tuft dendrites of CA1 pyramidal neurons (left) and z-scored iGluSnFR4f traces from denoted spines (right). Scale bar, 2 μm. **c)** Amplitudes of synaptic glutamate signals from basal (n = 265 ROIs from 4 cells), oblique (n = 286 ROIs from 4 cells), and tuft dendrites (n = 444 ROIs from 4 cells)

of pyramidal neurons in CA1 and layer 2/3 dendrites (n = 465 ROIs from 4 cells) in V1 Kruskal-Wallis test, P = 2.2×10-16. Basal vs Oblique: P = 6.2×10-2; Basal vs Tuft: P = 2.1×10-82; Oblique vs Tuft: P = 2.8×10-69; Basal vs V1: P = 2.2×10-13; Oblique vs V1: 5.6×10-8; Tuft vs V1: P = 9.6×10-43. **d)** Distribution of Inter-Event-Interval for glutamate signals from different dendritic domains of CA1 pyramidal neurons. Basal vs Oblique: P = 0.0073; Oblique vs Tuft: P = 4.9×10-24; Basal vs Tuft: P = 1.7×10-13.; Kolmogorov–Smirnov test.

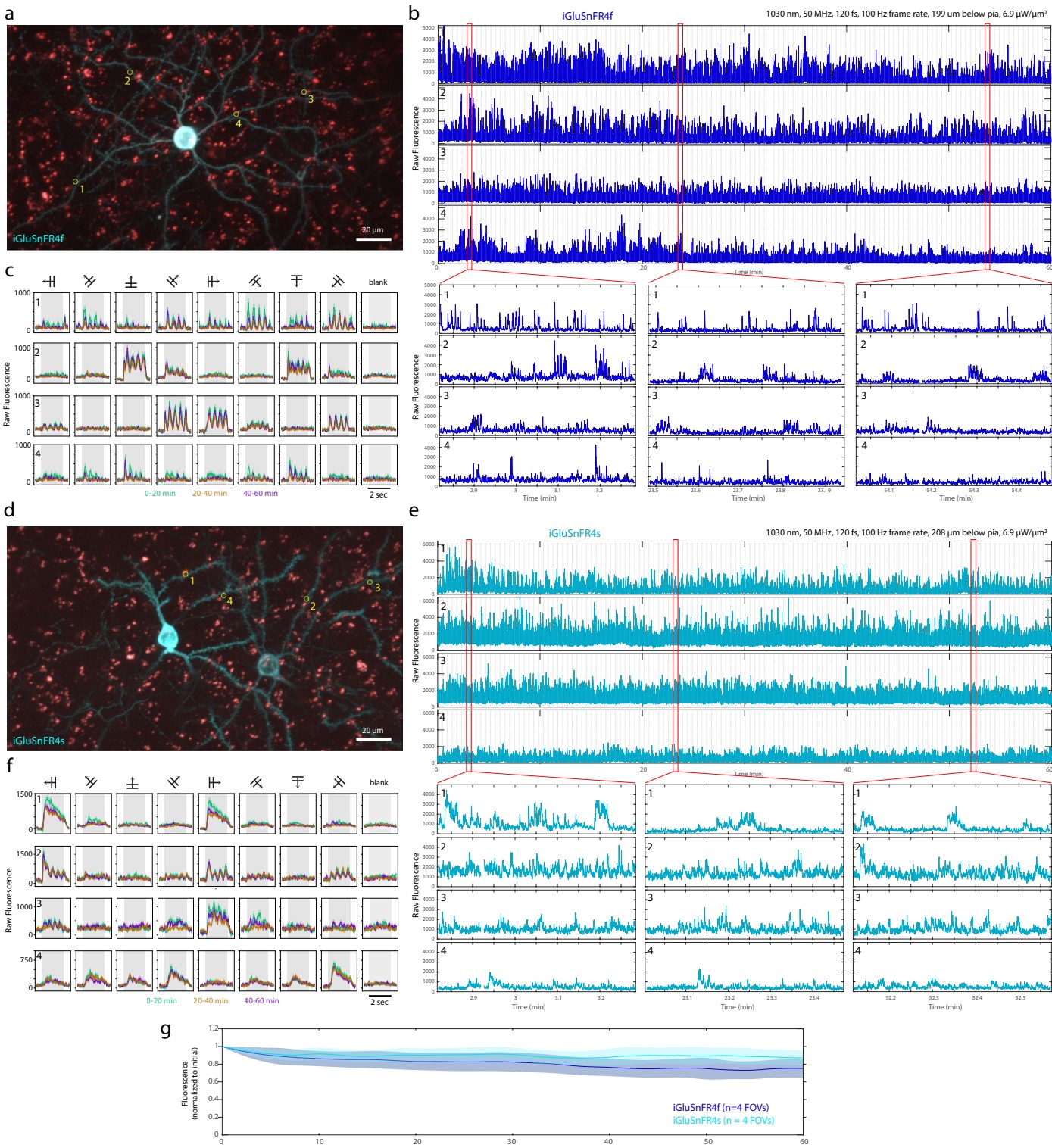

**Extended Data Fig. 8 | Long-duration continuous imaging of iGluSnFR4 in cortical dendrites.** Awake mice were shown drifting grating visual stimuli for 1 h while imaging dendrites at 100 Hz. **a)** Intensity projection (maximum over 20 μm depth) of example imaged neurons. Circles denote spines whose activity is plotted in b and c. Red puncta are endogenous lipofuscin. **b)** Top, Raw fluorescence traces (1 h at 100 Hz) recorded from the highlighted spines. Vertical lines denote boundaries between each block of the 9 presented stimuli. Bottom, zooms of individual stimulus blocks from throughout recording. **c)** Stimulus-triggered average raw fluorescence for the highlighted spines, for the first (cyan), second (orange) and third (purple) 20-minute periods in the 1-hour recording. Shading denotes SEM (N = 42-44 repeats per block). **(d-f)** same as **(a-c)**, for iGluSnFR4s. **g)** Low-pass filtered brightness (mean ± SEM) normalized to initial, for the two indicators. N = 4 FOVs each.

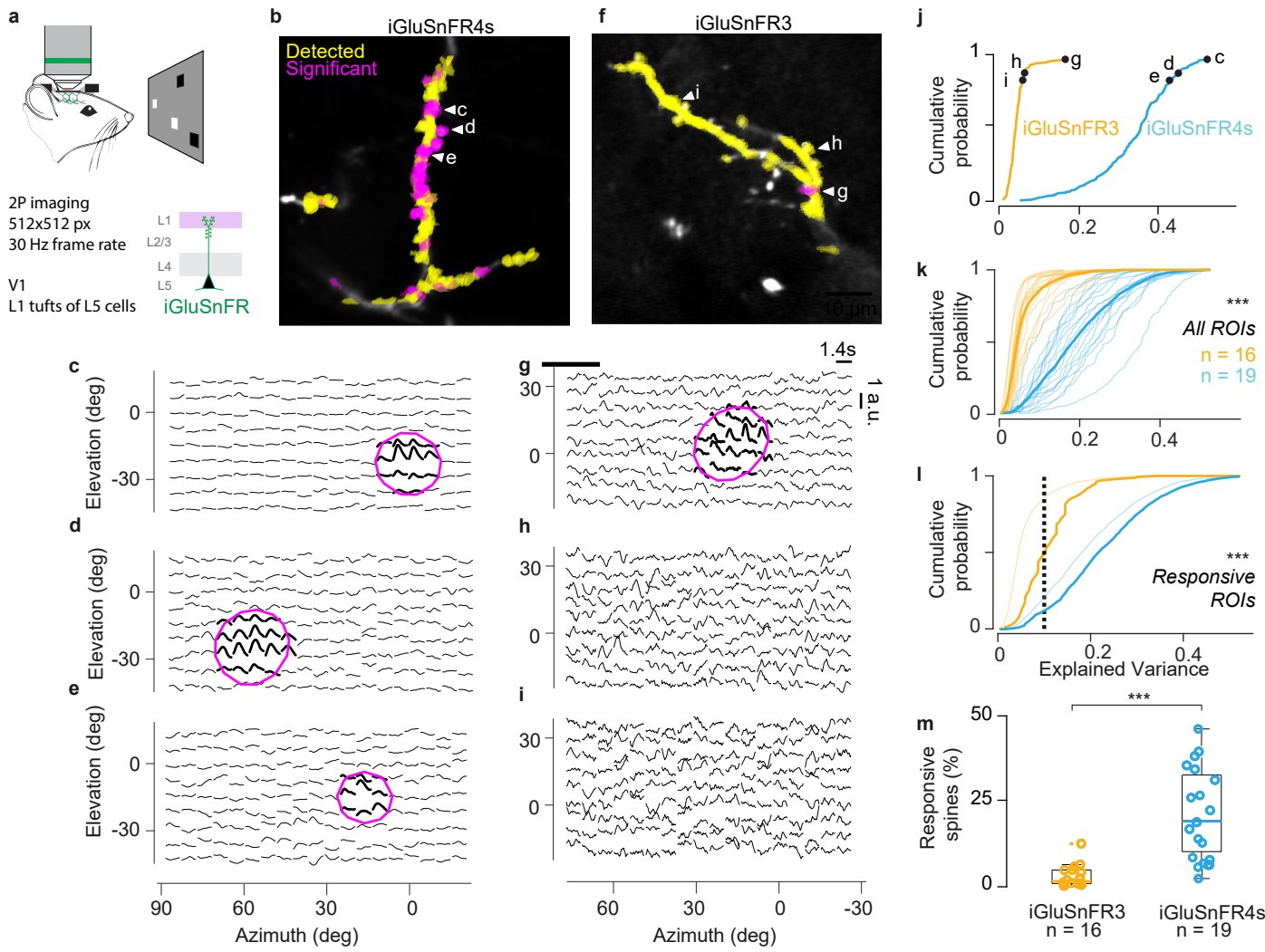

**Extended Data Fig. 9 | 30 Hz imaging in visual cortex. a)** Mice passively viewed sparse noise stimuli (sequences of black and white squares, each appearing for 200 ms in random locations) while imaging apical dendrites of L5 neurons in mouse V1 sparsely labeled with iGluSnFR3 (16 dendrites from 7 neurons in 3 mice) or iGluSnFR4s (19 dendrites, 3 neurons, 2 mice). Because dendrites or neurons are not statistically independent, we employed a linear mixed effect model to analyze results. **b)** Example mean image of an apical dendrite expressing iGluSnFR4s. Active ROIs with significant receptive fields (magenta; p < 0.005 vs circular shift null and >10% variance explained by stRF) and all others detected by Suite2p (yellow) are overlaid to the normalised mean fluorescence image. **c–e)** Spatiotemporal receptive fields (stRFs) fit for ROIs labelled in (b), normalized to peak response. Gaussian fit overlaid(magenta ellipse and thick traces). **f-i)** Same as (b-e) for example dendrite expressing iGluSnFR3. **j)** Cumulative probability distributions of the explained variance from all the ROIs detected

on the two example dendrites shown in b and f. The example stRF shown in c-e and g-i were randomly selected as those at the 85th, 90th and 100th percentile of the distribution (p = 9.9×10⁻⁸⁶, Kolmogorov-Smirnov test). **k)** Cumulative distribution of RF explained variance from all recorded dendrites (thin lines) and the average across dendrites (thick lines) for iGluSnFR3 (orange, 16 dendrites), and iGluSnFR4s (cyan, 19 dendrites). (p = 1.6×10⁻¹⁶, Kolmogorov-Smirnov test). **l)** Average cumulative probability distributions of the explained variance of all stRF (thin lines, same as k) or of stRF explaining more variance than chance ROIs only (thick line), for iGluSnFR3 (orange) and iGluSnFR4s (cyan). The dotted line shows the threshold of explained variance fraction (0.1) used to determine visual responsiveness. (p = 1.7×10⁻⁸⁰, Kolmogorov-Smirnov test). **m)** The percentage of responsive ROIs was significantly higher for dendrites expressing iGluSnFR4s (cyan) compared to iGluSnFR3 (orange, p = 1.7×10⁻⁶, linear mixed-effects model).

**Extended Data Table 1 | Photophysical characterization of iGluSnFR3, iGluSnFR4f and iGluSnFR4s**

| Variant Name | iGluSnFR3 | | iGluSnFR4f | | iGluSnFR4s | |
|---|---|---|---|---|---|---|
| | APO | SAT | APO | SAT | APO | SAT |
| 1-photon Excitation maxima $\lambda_{exc}$ (nm) | 502 | | 502 | | 502 | |
| 1-photon Emission maxima $\lambda_{exc}$ (nm) | 522 | | 522 | | 522 | |
| 1-photon ΔF/F | 47.1 | | 22.9 | | 17.2 | |
| $K_d$ (μM) [95% C.I.] | 383.8 μM [325.2, 453.0] | | 49.9 μM [44.5, 56.1] | | 7.0 μM [5.6, 8.7] | |
| Apparent Hill coefficient ($n_H$) [95% C.I.] | 0.9 [0.8, 1.1] | | 1.1 [1.0, 1.2] | | 1.0 [0.8, 1.2] | |
| Apparent pKa | 7.0 | 5.7 | 7.4 | 6.5 | 7.9 | 6.7 |
| Extinction Coefficient ε ($M^{-1}cm^{-1}$) | 2055 | 35000 | 3510 | 32920 | 4430 | 40440 |
| Quantum Yield Φ | 0.29 | 0.88 | 0.27 | 0.89 | 0.46 | 0.88 |
| 1-photon Brightness (x1000) | 0.60 | 30.8 | 0.95 | 29.3 | 2.0 | 35.6 |
| 2-photon Brightness $F_2$(GM), (950 nm / 1030 nm) | 18.8 / 6.4 | | 17.3 / 6.1 | | 20.7 / 7.2 | |
| 2-photon ΔF/F, (950 nm / 1030 nm) | 43.1 / 48.4 | | 30.6 / 35.7 | | 15.7 / 16.3 | |

# Reporting Summary

## Statistics

For all statistical analyses, confirm that the following items are present in the figure legend, table legend, main text, or Methods section.

| n/a | Confirmed | |
|---|---|---|
| ☐ | ☒ | The exact sample size (*n*) for each experimental group/condition, given as a discrete number and unit of measurement |
| ☐ | ☒ | A statement on whether measurements were taken from distinct samples or whether the same sample was measured repeatedly |
| ☐ | ☒ | The statistical test(s) used AND whether they are one- or two-sided<br>*Only common tests should be described solely by name; describe more complex techniques in the Methods section.* |
| ☐ | ☒ | A description of all covariates tested |
| ☐ | ☒ | A description of any assumptions or corrections, such as tests of normality and adjustment for multiple comparisons |
| ☐ | ☒ | A full description of the statistical parameters including central tendency (e.g. means) or other basic estimates (e.g. regression coefficient) AND variation (e.g. standard deviation) or associated estimates of uncertainty (e.g. confidence intervals) |
| ☐ | ☒ | For null hypothesis testing, the test statistic (e.g. *F*, *t*, *r*) with confidence intervals, effect sizes, degrees of freedom and *P* value noted<br>*Give P values as exact values whenever suitable.* |
| ☒ | ☐ | For Bayesian analysis, information on the choice of priors and Markov chain Monte Carlo settings |
| ☐ | ☒ | For hierarchical and complex designs, identification of the appropriate level for tests and full reporting of outcomes |
| ☐ | ☒ | Estimates of effect sizes (e.g. Cohen's *d*, Pearson's *r*), indicating how they were calculated |

*Our web collection on statistics for biologists contains articles on many of the points above.*

## Software and code

Policy information about availability of computer code

| | |
|---|---|
| Data collection | Custom MATLAB and Python scripts were used for high-throughput imaging analysis, electrophysiology, and fluorescence quantification. Software tools included MATLAB (versions 2021b through 2024b, as reported in the Methods), Python 3.9 (NumPy, SciPy, matplotlib), and ScanImage 2023 for microscope control. The GENIE project team used custom code for data acquisition (lightly modified from Wardill et al 2013; available from the GENIE project team upon request). Widefield imaging used WaveSurfer v1.0.6 and Hamamatsu HCImage 4.3.1, and NIS-Elements 4.1. |
| Data analysis | Data analysis was performed using custom Python and MATLAB code. GLM models were fitted using statsmodels (v0.13.5) in Python. Imaging data was processed with Suite2P and NMF-based algorithms. Code used for processing and analysing data is available in the code and data supplement (doi: 10.25378/janelia.30251743). |

For manuscripts utilizing custom algorithms or software that are central to the research but not yet described in published literature, software must be made available to editors and reviewers. We strongly encourage code deposition in a community repository (e.g. GitHub). See the Nature Portfolio guidelines for submitting code & software for further information.

## Data

Policy information about availability of data

All manuscripts must include a data availability statement. This statement should provide the following information, where applicable:
  - Accession codes, unique identifiers, or web links for publicly available datasets
  - A description of any restrictions on data availability
  - For clinical datasets or third party data, please ensure that the statement adheres to our policy

All plasmids, constructs, and sequences used in this study are publicly available on Addgene: https://www.addgene.org/browse/article/28252906/

Structural data and the iGluSnFR3-based model used for iGluSnFR4 engineering are available via the RCSB Protein Data Bank under accession code 9FBU. https://www.rcsb.org/structure/9FBU

## Research involving human participants, their data, or biological material

Policy information about studies with human participants or human data. See also policy information about sex, gender (identity/presentation), and sexual orientation and race, ethnicity and racism.

| | |
|---|---|
| Reporting on sex and gender | *Use the terms sex (biological attribute) and gender (shaped by social and cultural circumstances) carefully in order to avoid confusing both terms. Indicate if findings apply to only one sex or gender; describe whether sex and gender were considered in study design; whether sex and/or gender was determined based on self-reporting or assigned and methods used. Provide in the source data disaggregated sex and gender data, where this information has been collected, and if consent has been obtained for sharing of individual-level data; provide overall numbers in this Reporting Summary. Please state if this information has not been collected. Report sex- and gender-based analyses where performed, justify reasons for lack of sex- and gender-based analysis.* |
| Reporting on race, ethnicity, or other socially relevant groupings | *Please specify the socially constructed or socially relevant categorization variable(s) used in your manuscript and explain why they were used. Please note that such variables should not be used as proxies for other socially constructed/relevant variables (for example, race or ethnicity should not be used as a proxy for socioeconomic status). Provide clear definitions of the relevant terms used, how they were provided (by the participants/respondents, the researchers, or third parties), and the method(s) used to classify people into the different categories (e.g. self-report, census or administrative data, social media data, etc.) Please provide details about how you controlled for confounding variables in your analyses.* |
| Population characteristics | *Describe the covariate-relevant population characteristics of the human research participants (e.g. age, genotypic information, past and current diagnosis and treatment categories). If you filled out the behavioural & social sciences study design questions and have nothing to add here, write "See above."* |
| Recruitment | *Describe how participants were recruited. Outline any potential self-selection bias or other biases that may be present and how these are likely to impact results.* |
| Ethics oversight | *Identify the organization(s) that approved the study protocol.* |

Note that full information on the approval of the study protocol must also be provided in the manuscript.

# Field-specific reporting

Please select the one below that is the best fit for your research. If you are not sure, read the appropriate sections before making your selection.

☒ Life sciences  ☐ Behavioural & social sciences  ☐ Ecological, evolutionary & environmental sciences

For a reference copy of the document with all sections, see nature.com/documents/nr-reporting-summary-flat.pdf

# Life sciences study design

All studies must disclose on these points even when the disclosure is negative.

| | |
|---|---|
| Sample size | No statistical method was used to pre-determine sample size. Sample sizes were matched to prior work with iGluSnFR3 and other fluorescent indicators, which were previously sufficient to detect differences in key metrics such as $\Delta F/F_0$, signal-to-noise ratio (SNR), and kinetic parameters. All main results were observed consistently across replicates. |
| Data exclusions | Data were excluded from analysis only on the basis of technical failures during data acquisition. |
| Replication | All key findings, including in vitro and in vivo indicator performance, were successfully replicated in at least two independent biological replicates. No major findings failed replication. |
| Randomization | Samples and animals were randomized across conditions to minimize bias. For in vitro screening, iGluSnFR4 variants were distributed across wells, and the layout was randomized across plates to control for positional and batch effects. For in vivo imaging, mice were randomly assigned indicator constructs at the time of AAV injection. Imaging regions (e.g., dendrites or boutons) were selected without knowledge of |

expected outcomes. Randomization was maintained throughout data acquisition and analysis pipelines to ensure fair comparisons between variants.

**Blinding**

The only experiment that involved blinding was the manual annotation of spine survival. In all other cases, blinding was unnecessary because data collection used highly automated instruments and/or clearly defined protocols that minimize the risk of experimenter bias. The benefit of blinding would not warrant the additional complexity and potential errors associated with it in this context. Similarly, the analyses performed were virtually all automated, making blinding irrelevant. Manual annotation was necessary to identify synaptic structures in two-photon volumes; annotations were not performed blinded, but were subsequently assessed by an expert blinded to indicator identity before analysis. Examples shown in figures were either randomly selected or manually selected as representative examples typical of the experiment.

# Reporting for specific materials, systems and methods

We require information from authors about some types of materials, experimental systems and methods used in many studies. Here, indicate whether each material, system or method listed is relevant to your study. If you are not sure if a list item applies to your research, read the appropriate section before selecting a response.

## Materials & experimental systems

| n/a | Involved in the study |
|-----|----------------------|
| ☒ | Antibodies |
| ☒ | Eukaryotic cell lines |
| ☒ | Palaeontology and archaeology |
| ☐ | ☒ Animals and other organisms |
| ☒ | Clinical data |
| ☒ | Dual use research of concern |
| ☒ | Plants |

## Methods

| n/a | Involved in the study |
|-----|----------------------|
| ☒ | ChIP-seq |
| ☒ | Flow cytometry |
| ☒ | MRI-based neuroimaging |

## Animals and other research organisms

Policy information about studies involving animals; ARRIVE guidelines recommended for reporting animal research, and Sex and Gender in Research

**Laboratory animals**

Mice: Emx1-cre (JAX 005628), female, 8–30 wks; C57BL/6, mixed sexes, 8–30 wks; Scnn1a-Tg3-Cre (JAX 009613), male, 8–30 wks

**Wild animals**

The study did not involve wild animals.

**Reporting on sex**

Both male and female mice were used in this study. No sex-specific differences were observed in results, and data from both sexes were pooled.

**Field-collected samples**

The study did not involve field-collected samples.

**Ethics oversight**

All animal procedures were conducted in accordance with institutional animal care and use protocols approved by the Allen Institute for Neural Dynamics, Janelia Research Campus of HHMI, the University of California San Diego, Technical University of Munich, University College London, and St. Jude Children's Research Hospital. All procedures followed the relevant national and institutional guidelines for the care and use of laboratory animals.

Allen Institute for Neural Dynamics: Protocol 2109
HHMI Janelia Research Campus: Protocols 19-176, 22-0214.01
University of California, San Diego: Protocol S02174M
Technical University of Munich: Protocols 2532.Vet_02-23-24 and 2532.Vet_02-21-121
University College London: Animal License PP3929312
St. Jude Children's Research Hospital: Protocol no. 3193

Note that full information on the approval of the study protocol must also be provided in the manuscript.

# Plants

Seed stocks

*Report on the source of all seed stocks or other plant material used. If applicable, state the seed stock centre and catalogue number. If plant specimens were collected from the field, describe the collection location, date and sampling procedures.*

Novel plant genotypes

*Describe the methods by which all novel plant genotypes were produced. This includes those generated by transgenic approaches, gene editing, chemical/radiation-based mutagenesis and hybridization. For transgenic lines, describe the transformation method, the number of independent lines analyzed and the generation upon which experiments were performed. For gene-edited lines, describe the editor used, the endogenous sequence targeted for editing, the targeting guide RNA sequence (if applicable) and how the editor was applied.*

Authentication

*Describe any authentication procedures for each seed stock used or novel genotype generated. Describe any experiments used to assess the effect of a mutation and, where applicable, how potential secondary effects (e.g. second site T-DNA insertions, mosiacism, off-target gene editing) were examined.*

