## [Peer Review File · Nature Methods]

Glutamate indicators with increased sensitivity and tailored deactivation rates

Corresponding Author: Dr Kaspar Podgorski

Version 0:

Decision Letter:

16th May 2025

Dear Kaspar,

Your Article, "Glutamate indicators with increased sensitivity and tailored deactivation rates", has now been seen by 3 reviewers. As you will see from their comments below, although the reviewers find your work of considerable potential interest, they have raised a number of concerns. We are interested in the possibility of publishing your paper in Nature Methods, but would like to consider your response to these concerns before we reach a final decision on publication. We therefore invite you to revise your manuscript to address these concerns.

Link Redacted

We hope to receive your revised paper within 2-3 months. If you cannot send it within this time, please let us know. In this event, we will still be happy to reconsider your paper at a later date so long as nothing similar has been accepted for publication at Nature Methods or published elsewhere.

OPEN SCIENCE REQUIREMENTS

REPORTING SUMMARY AND EDITORIAL POLICY CHECKLISTS

EXTENDED DATA FIGURES

DATA AVAILABILITY

All novel DNA and RNA sequencing data, protein sequences, genetic polymorphisms, linked genotype and phenotype data, gene expression data, macromolecular structures, and proteomics data must be deposited in a publicly accessible database, and accession codes and associated hyperlinks must be provided in the "Data Availability" section.

MATERIALS AVAILABILITY

Authors reporting new chemical compounds must provide chemical structure, synthesis and characterization details. Authors

reporting mutant strains and cell lines are strongly encouraged to use established public repositories.

ORCID

Nature Methods is committed to improving transparency in authorship. As part of our efforts in this direction, we are now requesting that all authors identified as 'corresponding author' on published papers create and link their Open Researcher and Contributor Identifier (ORCID) with their account on the Manuscript Tracking System (MTS), prior to acceptance. This applies to primary research papers only. ORCID helps the scientific community achieve unambiguous attribution of all scholarly contributions. You can create and link your ORCID from the home page of the MTS by clicking on 'Modify my Springer Nature account'. For more information please visit <http://www.springernature.com/orcid>.

Best regards,
Nina

Nina Vogt, PhD
Senior Editor
Nature Methods

Reviewers' Comments:

Reviewer #1 (Remarks to the Author):

The current MS describes the development of two new genetically encoded glutamate sensors (iGluSnFR4s and f). The authors have already made important contributions to this field by developing previous versions of glutamate sensors. Here, they report extensive work, demonstrating further improvements. The MS contains a large body of in vitro and in vivo data, demonstrating improved sensitivity, SNR, stability of the new sensors and their usefulness in neuroscience. The new sensors will be clearly useful for researchers in the field. There are some inconsistencies within the MS and some missing quantitative data should be provided.

One of the most important features of such sensors is the photostability (apart from the high SNR, of course). The authors mentioned in the text that they measured this, but they show no (very little in the Suppl) data. It would be extremely useful for the readers and future users to see quantitative data on this important feature. The reviewer is aware of the fact that it depends on imaging conditions, microscopes etc, but if the authors can perform stable recordings for e.g. 20 min without detectable photobleaching, that would serve as a benchmark for the new users.

Another important feature of these reporters is their 'linearity' (or lack of it). How do the new sensors report the simultaneous release of two, three etc synaptic vesicles from the same synapse? This can be tested by imaging evoked responses at individual spine level and changing release probability. Would the expression level change the linearity?

Fig 1f, g show that single AP-evoked responses have amplitudes of ~3 and 5 dF/F with the new sensors. In Fig. 3d, these values are around 1! In Fig. 4i, they are 0.2-0.4! Here, the authors might argue that it is the consequence of averaging lots of failures to the signal, but the raw data does not support that hypothesis. The reviewer cannot see any release failures! How do the authors explain this?

The authors aimed to characterize the spatial crosstalk between spines with drifting grating stimuli. The analysis relies on the natural occurrence of neighbouring spines. Here the reviewer would like to point out that if the authors do not see all spines, only the large ones, then they underestimate the real cross-talk between neighbouring spines. It is widely accepted that spine density measurements in pyramidal cells are largely underestimated with 2P microscopy. The real density can be reliably measured with EM and even STED superresolution microscopy underestimate the real density. The distances between the marked spines in Fig. 3g are >2 μ m. The density estimate from these 2P in vivo images is <1 / μ m dendritic length. That is much lower than the real spine density!

Why do fluorescent transients have rise times around few ms and also tens of ms (fig. 1g).

In Fig. 2C the Y scale bar is the same for all traces. The baseline noise is clearly very different (e.g. third dark blue trace)! Why?

The section describing video rate imaging in the visual cortex is very weakly supported by data and is not very conclusive.

How was the SNR calculated for Fig. 2D? The amplitude of the responses show large variability, therefore the SNR varies tremendously from mini to mini? How do the authors explain such a large variability in amplitude if they assume that all of the minis are the consequence of the release of a single vesicle?

The measurements of the duration is 3-4-fold faster for iGluSnFR4f (Fig. 2h) whereas the traces presented in panel G look very similar. Why?

Last part of the result section: 'iGluSnFR4s produced responses with larger amplitude than iGluSnFR3 (5.62 ± 0.42 vs 1.56 ± 0.16) or SF-iGluSnFR (3.72 ± 0.92 vs 1.54 ± 0.44) in the paired measurements (Fig. 5c-d).' Why iGluSnFR4s has different amplitudes when compared to iGluSnFR3 vs SF-iGluSnFR? If the response amplitudes can be so variable, what does it mean? These experiments add very little to the MS.

The text has many abbreviations without spelling out the full names (e.g. NGR and PDGFR).

Reviewer #2 (Remarks to the Author):

Glutamatergic neurotransmission is a fundamental process underlying information processing and transmission in the nervous system. Simultaneous recording of glutamatergic neurotransmission at a large number of synapses in vivo is thus essential for an understanding of the cellular principles of neural computation. However, this is technically challenging to accomplish. Aggarwal et al. screened a large library of rationally-targeted mutations in the genetically encoded fluorescence glutamate indicator iGluSnFR3 to develop two new sensors, iGluSnFR4s/f, with improved sensitivity and brightness, and validated them in several systems. This work provides a valuable tool for cellular and systems neuroscience research. However, some questions and concerns need to be clarified.

1. Fig. 1f and g shows that iGluSnFR4s and f have higher dF/F_0 in response to 1 AP than iGluSnFR3. Fig. 2g and Fig. 3b-d show that a consistent trend. However, Supplementary Table 1 suggests that the 1P and 2P dF/F_0 of iGluSnFR4s/f are lower than that of iGluSnFR3. How were the values given in Suppl. Table 1 measured? How should we understand this?
2. The iGluSnFRs are attached to the cell membrane by the GPI anchor. Are they targeted to synaptic sites specifically? If so, how?
3. Supplementary Fig. 7b characterizes the photostability of different iGluSnFR variants. It shows that iGluSnFR4s (cyan) intensity decays slower than iGluSnFR3/4f. However, figure legend states that 4f has a much larger tau than the other two variants. Is there a typo? Which iGluSnFR4 variant is more photostable?
4. Fig. 3a-d imaged iGluSnFR signals in axons, supposedly at presynaptic boutons. How should we understand such axonal signals? Given that cortical synapses are generally unreliable, one would expect no glutamate signal in response to a significant proportion of single APs, which does not seem to be the case in Fig. 3b. Overall, what is the failure rate of synaptic transmission measured by glutamate imaging in Fig. 3b? What is the amplitude distribution of single AP-triggered events?

Reviewer #3 (Remarks to the Author):

Glutamate is one of the most important signaling molecules, playing a dominant role in neural transmission in the vertebrate brain. To monitor glutamate dynamics in vivo, genetically encoded glutamate sensors have been developed and have become valuable tools for studying synaptic transmission and neural circuits. In this work, the authors aim to improve the SNR of glutamate sensors through structure-guided engineering and extensive screening. Following screening under field stimulation, optical minis, and in vivo visual stimuli, two top-performing variants—iGluSnFR4f and iGluSnFR4s—were identified, both exhibiting significantly enhanced SNR compared to iGluSnFR3. The authors further characterized these sensors and demonstrated their performance across multiple brain regions and imaging modalities, including one-photon and two-photon microscopy, as well as fiber photometry. These newly developed sensors represent a significant advancement and are expected to be highly valuable for the neuroscience community, facilitating studies of synaptic communication.

Major comments:

1. While the improvement in iGluSnFR4 is appreciated, the raw traces of iGluSnFR3 shown in Fig. 2c appear highly skewed—the first ROI shows a strong response, whereas the others are relatively weak. It is unclear whether this discrepancy is due to random sampling or reflects the limited sensitivity of iGluSnFR3. Including an additional panel quantifying the cumulative responses or the number of above-threshold events for all three variants would better highlight the superiority of the fourth-generation sensors.
2. In Fig. 3i, the authors chose to use dF/F_0 for comparison. However, as shown in Fig. 2h, iGluSnFR4f has a substantially lower F_0 than iGluSnFR4s, while the SNRs between them are similar. Including a direct comparison of SNR in Fig. 3i would provide a more comprehensive evaluation of sensor performance.
3. Including a mutant sensor that does not bind glutamate or lacks a fluorescence response to glutamate would be highly beneficial for users, especially as a control in fiber photometry experiments. This would help rule out non-specific signals and enhance the interpretability and rigor of future studies.

Minor comments:

1. I recommend including raw data points in Fig. 1g along with statistical analysis. Given the large sample sizes, t-tests or ANOVA may be overly sensitive. Non-parametric tests such as the Kolmogorov–Smirnov (K-S) test or the Mann–Whitney U

test may provide a fairer assessment. The same recommendation applies to Fig. 2h.

2. In Fig. 2h, the labeling of sensor variants is somewhat confusing: “v8880” and “v9601” correspond to iGluSnFR4s and iGluSnFR4f, which were already defined earlier before this panel. The authors may consider unifying the labeling to maintain consistency throughout.

3. Unlike most green sensors such as GCaMP, which primarily exhibit activation-dependent changes in absorbance, iGluSnFR4s/f show changes in both absorbance and quantum yield upon glutamate binding. It would be valuable if the authors could provide any structural insights or mechanistic hypotheses that might explain this distinctive property.

4. In Fig. 3b, there appears to be some inconsistency between the cell voltage and the iGluSnFR3 response, whereas the responses of iGluSnFR4s/f sensors correlate more closely with voltage changes. It is unclear whether this apparent “fatigue” in iGluSnFR3 reflects lower sensor sensitivity or is related to data quality. Additionally, the firing rate in the iGluSnFR3 trace seems notably lower than in the iGluSnFR4s/f traces. The authors might consider replacing this example with more representative data to avoid potential confusion.

5. I appreciate the authors’ detailed analysis of the detectability for iGluSnFR3 and iGluSnFR4 sensors presented in Supplementary Figs. 9 and 10. I encourage the authors to include some of this analysis in the main figures to better highlight the performance improvements of iGluSnFR4.

6. Supplementary Fig. 7 compares photobleaching properties of iGluSnFR3 and iGluSnFR4s/f. Extending this comparison to include parent fluorescent proteins or commonly used sensors such as GCaMP would be informative.

7. The scale bar label in Fig. 1d should be μM not μM .

Version 1:

Decision Letter:

Our ref: NMETH-A60266A

4th Sep 2025

Dear Kaspar,

Thank you for submitting your revised manuscript “Glutamate indicators with increased sensitivity and tailored deactivation rates” (NMETH-A60266A). As I mentioned already, it has been seen by the original referees and their comments are below. The reviewers find that the paper has improved in revision. Taking into account the additional information you provided, we'll be happy in principle to publish your manuscript in Nature Methods, pending minor revisions to satisfy the referees' final requests and to comply with our editorial and formatting guidelines.

Please ensure that the additional explanations you provided will be included in the manuscript.

I would also like to ask whether I can forward your response to the reviewers' concerns to reviewer #1 as a courtesy.

TRANSPARENT PEER REVIEW

Nature Methods offers a transparent peer review option for new original research manuscripts. We encourage increased transparency in peer review by publishing the reviewer comments, author rebuttal letters and editorial decision letters if the authors agree. Such peer review material is made available as a supplementary peer review file. **Please state in the cover letter ‘I wish to participate in transparent peer review’ if you want to opt in, or ‘I do not wish to participate in transparent peer review’ if you don’t.** Failure to state your preference will result in delays in accepting your manuscript for publication.

Please note: we allow redactions to authors’ rebuttal and reviewer comments in the interest of confidentiality. If you are concerned about the release of confidential data, please let us know specifically what information you would like to have removed. Please note that we cannot incorporate redactions for any other reasons. Reviewer names will be published in the peer review files if the reviewer signed the comments to authors, or if reviewers explicitly agree to release their name. For more information, please refer to our <https://www.nature.com/documents/nr-transparent-peer-review.pdf> target="new">FAQ page.

ORCID

Best regards,
Nina

Nina Vogt, PhD
Senior Editor
Nature Methods

Reviewer #1 (Remarks to the Author):

Bleaching information for all of the variants in the in vivo screen, including iGluSnFR4f and 4s, was included in main Figure 2h ("Bleaching Fraction"; now in panel 2i). Note that the bleaching in those recordings was more pronounced than in our long-term recordings because they were performed at very high illumination power, which supralinearly accelerates bleaching (Eggeling, Volkmer, and Seidel 2005; Gavriluk et al. 2007). In these conditions, iGluSnFR4f showed greater and iGluSnFR4s lower photostability than iGluSnFR3 (Aggarwal et al. 2023), whereas the longer, low power recordings show the opposite trend. Despite this apparent power-dependence, Suppl. Figure 10 shows that long-term recordings are possible with both indicators at excellent SNR.

The reviewer cannot see these in Figure 2h and 2i.

1) (lines 222-225) Synaptic inputs to V1 neurons have diverse preferences for grating orientation, direction, and phase, exhibiting little spatial organization over micrometer scales. Spatial crosstalk would blend signals with different tuning, reducing measured selectivity. Both iGluSnFR4f and 4s reported high signal-to-noise, orientation-tuned responses localized to dendritic spines. We observed clear responses for each cycle of the grating stimuli from many distinct sites.

Expanded explanation:

Because different axons respond to the grating stimuli with different stimulus preferences, crosstalk would cause us to observe mixed selectivity, and we would not, for example, see such clear phasic responses to specific stimulus orientations. This is true regardless of the density of synapses we identify

That is clearly true if there was absolutely no response to different orientations, only to the preferred one. However, this is clearly not the case (see fig 3!). Many spines have responses to the non-preferred orientation as well. This could be biological, because the input is not tuned only to a single orientation, or could be the consequence of crosstalk from neighbouring synapses. It is impossible to distinguish between these two possibilities.

Rise times in the field stimulation screen (Fig 1g) are slower than in vivo (Fig 3) for several reasons:

- 1) Glutamate diffuses from release sites and contacts much of the labeled membrane only milliseconds after release. Our analysis of field stimulation data includes all responsive pixels so it captures this effect.
- 2) Field stimulation causes release from many boutons at once. The fact that many release sites are stimulated simultaneously increases the effective length scale of diffusion; the time scale of diffusion is proportional to the square of the length scale.
- 3) Glial reuptake may differ in the cultures versus in vivo, interacting with the above two effects and otherwise directly affecting both the length- and time- scales of signals.
- 4) These culture experiments were performed at room temperature, resulting in slower kinetics than in vivo experiments.

The reviewer does not agree with this. The diffusion of glutamate within the cleft is estimated to be around 100 microseconds. Binding of glutamate to transporters is a very slow process, therefore it does not contribute to the rise time of the postsynaptic responses. The on-rate of AMPA receptors is not very different from that of the GluSnFR, and therefore the optical signal can have fast rise times. The slow rise must indicate, therefore, binding to the reporters at large distances (spillover). If this happens upon some conditions, then the synapse specificity is collapsed (see previous point). This should be acknowledged.

In Fig. 2C the Y scale bar is the same for all traces. The baseline noise is clearly very different (e.g. third dark blue trace)! Why?

The traces are plotted normalized to their standard deviations (not by an estimate of the baseline noise). We have double-checked and there are no bugs or errors in how this scaling was performed. For the purpose here, of showing example traces, we consider this simple scaling appropriate.

The reviewer acknowledges that there was no error. The question, however, remains: why is the background noise so different between traces?

Last part of the result section: 'iGluSnFR4s produced responses with larger amplitude than iGluSnFR3 (5.62 ± 0.42 vs 1.56 ± 0.16) or SF-iGluSnFR (3.72 ± 0.92 vs 1.54 ± 0.44) in the paired measurements (Fig. 5c-d).' Why iGluSnFR4s has different amplitudes when compared to iGluSnFR3 vs SF-iGluSnFR? If the response amplitudes can be so variable, what does it mean? These experiments add very little to the MS. In these in vivo measurements, per-animal factors

The reviewer believes that the authors did not provide an adequate response to the issue. The argument of reward and neuromodulation etc might be true, but if the two sets of experiments are identical, then an identical portion of the recording should be under high expectation/reward/motivation etc therefore the baseline data should be the same. That is very alarming if the same sensor is used in two comparisons and the results with the same sensor under similar experimental conditions are highly different.

Reviewer #2 (Remarks to the Author):

The revision addresses all my questions. I don't have any further issue with it.

Reviewer #3 (Remarks to the Author):

The authors have done an excellent job of revising the manuscript, thoughtfully addressing the previously raised comments. The changes have significantly improved the clarity, depth, and overall quality of the article, thus more comprehensively demonstrated the importance and the superiority of iGluSnFR4. I am satisfied with the revisions and believe the manuscript is now ready for publication in its current form.
